# Chamber-based insights into the factors controlling IEPOX SOA yield, composition, and volatility

Emma L. D'Ambro[1,2], Siegfried Schobesberger[1,3], Cassandra J. Gaston[1,a], Felipe D. Lopez-Hilfiker[1,b], Ben H. Lee[1], Jiumeng Liu[4,c], Alla Zelenyuk[5], David Bell[5,d], Christopher D. Cappa[6,7], Taylor Helgestad[6,e], Ziyue Li[7], Alex Guenther[5,f], Jian Wang[8,g], Matthew Wise[9,h], Ryan Caylor[9], Jason D. Surratt[10], Theran Riedel[10], Noora Hyttinen[11,i], Vili-Taneli Salo[11], Galib Hasan[11], Theo Kurtén[11], John E. Shilling[4], Joel A. Thornton[1,2]

[1]Department of Atmospheric Sciences, University of Washington, Seattle WA, USA
[2]Department of Chemistry, University of Washington, Seattle WA, USA
[3]Department of Applied Physics, University of Eastern Finland, Kuopio, Finland
[4]Atmospheric Sciences and Global Change Division, Pacific Northwest National Laboratory, Richland WA, USA
[5]Environmental Molecular Sciences Laboratory, Pacific Northwest National Laboratory, Richland WA, USA
[6]Department of Civil and Environmental Engineering, University of California, Davis CA, USA
[7]Atmospheric Science Graduate Group, University of California, Davis CA, USA
[8]Environmental and Climate Sciences Department, Brookhaven National Laboratory, Upton NY, USA
[9]Department of Math and Science, Concordia University, Portland OR, USA
[10]Department of Environmental Sciences and Engineering, Gillings School of Global and Public Health, University of North Carolina, Chapel Hill NC, USA
[11]Department of Chemistry and Institute for Atmospheric and Earth System Research (INAR), University of Helsinki, Helsinki, Finland
[a]now at: Rosenstiel School of Marine & Atmospheric Science, University of Miami FL, USA
[b]now at: TofWerk AG, Thun, Switzerland
[c]now at: School of Environment, Harbin Institute of Technology, Harbin, Heilongjiang, China
[d]now at: Laboratory of Atmospheric Chemistry, Paul Scherrer Institute, PSI-Villigen, Switzerland
[e]now at: California Air Resources Board, Sacramento CA, USA
[f]now at: Department of Earth System Science, University of California, Irvine, USA
[g]now at: Center for Aerosol Science and Engineering, Department of Energy, Environmental and Chemical Engineering, Washington University in St. Louis, St. Louis MO, USA
[h]now at: Department of Chemistry, University of Colorado, Boulder CO, USA
[i]now at: Nano and Molecular Systems Research Unit, University of Oulu, Oulu, Finland

**Abstract**

We present measurements utilizing the Filter Inlet for Gases and Aerosols (FIGAERO) applied to chamber measurements of isoprene-derived epoxydiol (IEPOX) reactive uptake to aqueous acidic particles and associated SOA formation. Similar to recent field observations with the same instrument, we detect two molecular components desorbing from the IEPOX SOA in high abundance: $C_5H_{12}O_4$ and $C_5H_{10}O_3$. The thermal desorption signal of the former, presumably 2-methyltetrols, exhibits two distinct maxima, suggesting it arises from at least two different SOA components with significantly different effective volatilities. Isothermal evaporation experiments illustrate that the most abundant component giving rise to $C_5H_{12}O_4$ is semi-volatile, undergoing nearly complete evaporation within 1 hour, while the second, less volatile, component remains unperturbed and even increases in abundance. We thus confirm, using controlled laboratory studies, recent analyses of ambient SOA measurements showing that IEPOX SOA is of very low volatility and commonly measured IEPOX SOA tracers such as $C_5H_{12}O_4$ and $C_5H_{10}O_3$, presumably 2-methyltetrols and $C_5$-alkene triols or 3-MeTHF-3,4-diols, result predominantly from thermal decomposition in the FIGAERO-CIMS. We infer that other measurement techniques using thermal desorption or prolonged heating for analysis of SOA components may also lead to reported 2-methyltetrols and $C_5$-alkene triols or 3-MeTHF-3,4-diol structures.. We further show that IEPOX SOA volatility continues to evolve via acidity enhanced accretion chemistry on the timescale of hours, potentially involving both 2-methyltetrols and organosulfates.

**Introduction**

Aerosols less than 1 μm in diameter play particularly important roles in the Earth's radiative balance and air quality, a large fraction of which is organic carbonaceous material of biogenic origin (~70%) (Hallquist et al., 2009). Isoprene, with a global emission rate of 500 TgC year[-1] (Guenther et al., 2012), has the potential to form significant quantities of secondary organic aerosol (SOA). The ability of a volatile organic compound (VOC) to form SOA depends on either of two factors: the efficiency of its oxidative conversion to lower volatility products that can partition to the condensed phase, or the reaction of gas-phase oxidation products in the condensed phase to form products which remain in the condensed phase. Each of these two SOA formation mechanisms have been heavily studied in the case of isoprene.

The atmospheric oxidation of isoprene under low NO conditions typically proceeds by OH radicals, leading to the formation of first generation isoprene hydroxy hydroperoxides (ISOPOOH) in high yield (70%) (Paulot et al., 2009). The remaining double bond of isoprene can then undergo another OH addition, leaving a carbon-centered radical adjacent to a hydroperoxide moiety. This radical can internally rearrange to form an epoxy diol (IEPOX) at a yield of ~70-80% (St Clair et al., 2016) or undergo addition of $O_2$ to form a peroxy radical. The peroxy radical undergoes bimolecular reactions to form closed shell hydroperoxide or nitrate products, or unimolecular H-shifts to form carbonyl- and epoxide-containing products (Paulot et al., 2009; D'Ambro et al., 2017b). The bimolecular peroxy radical reaction products have been shown to be of sufficiently low volatility to partition to the aerosol-phase (D'Ambro et al., 2017a; Liu et al., 2016), and IEPOX has been shown to react in aqueous acidic particles, forming SOA. Commonly measured species from IEPOX reactive uptake include 2-methyltetrols (Lin et al., 2012; Surratt et al., 2006; Wang et al., 2005; Surratt et al., 2010), $C_5$-alkene triols (Lin et al., 2012; Surratt et al., 2010; Surratt et al., 2006; Wang et al., 2005), organosulfates (Lin et al., 2012; Surratt et al., 2010; Surratt et al., 2007b; Surratt et al., 2007a; Riva et al., 2019), 3-

methyltetrahydrofuran-3,4-diols (3-MeTHF-3,4-diols) (Lin et al., 2012), and oligomers (Lin et
al., 2012; Surratt et al., 2010; Lin et al., 2014).

Recently, studies have called into question whether these commonly measured
monomeric products of IEPOX multiphase chemistry exist in the particle-phase as measured
(Isaacman-VanWertz et al., 2016; Hu et al., 2016). We showed previously that components of
organic aerosol in the Southeast U.S. with compositions of $C_5H_{10}O_3$ and $C_5H_{12}O_4$ desorbed at
much higher temperatures, and therefore much lower volatilities, than would be expected based
on their composition (Lopez-Hilfiker et al., 2016). We concluded that the measured
compositions arise from lower volatility material in the SOA that thermally decomposes rather
than evaporating as the native species, and that, based on the relative abundance of these lower
volatility components, IEPOX SOA as a whole is typically comprised of very low volatility
material. When IEPOX was reacted in bulk solutions and analyzed via nuclear magnetic
resonance, no evidence was found for the production of $C_5$-alkene triols or 3-MeTHF-3,4-diols,
and although a second isomer of the MeTHF diol was observed (3-MeTHF-2,4-diols), the
formation rate from IEPOX was calculated to be so slow relative to nucleophilic addition that its
formation would be limited to situations where the aerosol had low water content (Watanabe et
al., 2018). These findings were further corroborated by comparing a novel chromatography
technique that does not involve heating to traditional GC/EI-MS analysis of IEPOX SOA
compositions, finding that alkene triols and 3-MeTHF-3,4-diols were in fact formed via thermal
decomposition of 2-methyltetrol sulfates and 3-methyltetrol sulfates, respectively (Cui et al.,
2018), and that organosulfates contribute significantly to the particle-phase (Riva et al., 2019).

An additional challenge to understanding IEPOX SOA is that there remains a large gap in
carbon closure resulting from IEPOX reactive uptake to aqueous acidic aerosol particles.
Although the reactivity of IEPOX in acidic particles is high (Gaston et al., 2014), the SOA yield
per reactive loss of IEPOX to particles is relatively low and varies greatly depending on aerosol
composition, from approximately 3 to 21% (Riedel et al., 2015). This disconnect may present an
inconsistency in models that simulate both IEPOX and IEPOX-derived SOA. If such low yields
are indeed realistic, models that adjust the rate of IEPOX reactive uptake to match the IEPOX
SOA tracer concentrations without accounting for the lower yield may not correctly simulate
IEPOX distributions where reactive uptake is a dominant sink for IEPOX (Gaston et al., 2014).

In this work we seek to understand the nature of products formed via the reactive uptake
of IEPOX in aqueous acidic particles and the flow of carbon between the gas- and particle-
phases. We present measurements from the Pacific Northwest National Laboratory (PNNL,
Richland WA) environmental chamber during the 2015 SOA Formation from Forest Emissions
Experiment (SOAFFEE) campaign. Batch and continuous flow mode experiments were
performed with authentic *trans*-β-IEPOX, which is the dominant isomer (Bates et al., 2014), and
wet acidic seed to study the products of IEPOX uptake and the resulting aerosol properties. The
properties of commonly measured IEPOX uptake products, $C_5H_{12}O_4$ (presumably 2-
methyltetrols) and $C_5H_{10}O_3$ (presumably $C_5$-alkene triols or 3-MeTHF-3,4-diols), such as
volatility and solubility, are examined in the context of experiments utilizing isothermal
evaporation of the formed SOA. The effect of acidity versus liquid water content on the products
formed is also discussed, along with implications for modeling atmospheric IEPOX and its
conversion into SOA.
**Experimental Methods**

The data presented herein were taken at the Pacific Northwest National Laboratory (PNNL) as part of the Secondary Organic Aerosol Formation from Forest Emissions Experiments (SOAFFEE) campaign held during the summer of 2015. PNNL's 10.6 m$^3$ fluorinated ethylene propylene (FEP) environmental chamber has been described in detail previously (Liu et al., 2012). The chamber was run in both batch mode and continuous flow mode. In batch mode the experiments lasted ~10 hours, while in continuous flow mode the total flow rate was 48.2 L min$^{-1}$ resulting in a residence time of ~3.7 hours.

Aliquots of an authentic *trans*-β-IEPOX standard which was synthesized according to Zhang et al. (2012) and dissolved in ethyl acetate were injected into a glass bulb which was connected to the chamber via ~10 cm of ¼" OD polytetrafluoroethylene (PTFE) tubing. Enough IEPOX was added to the empty chamber, before seed addition, to achieve 2 ppb in steady state and at the beginning of batch modes. The bulb and transfer line were heated to 30-40 $^o$C and a 100-300 sccm flow of zero air passed over the IEPOX to vaporize and carry it into the chamber. Typically, we measured ~ 10 µg m$^{-3}$ of IEPOX prior to aqueous seed addition. Given calibration uncertainties of ~ +/- 30% from repetitive injections and analytical errors, and potential losses on chamber and sampling surfaces, we estimate an uncertainty of approximately +/- 50% for gas phase IEPOX concentrations and other tracers. Wet, polydisperse acidic ammonium sulfate seed was generated by atomizing an ammonium bisulfate solution acidified with additional H$_2$SO$_4$. The solution was made by mixing ammonium sulfate (0.1308 g) with sulfuric acid (8.02 mL of 0.2465 M) and diluting to a total volume of 1 L with ultrapure water. The average molar NH$_4^+$:SO$_4^{2-}$ ratio measured by the AMS was approximately 0.93 for all experiments, though due to the experimental procedure some interference from organic sulfate formation may exist. The measured NH$_4^+$:SO$_4^{2-}$ ratio is significantly higher than was present in the atomized solution, implying that excess gas-phase ammonia present in the chamber partially neutralized the injected seed. The seed surface area concentrations were approximately 37,600 and 24,000-27,000 cm$^{-3}$ and the volume weighted mode diameters were 106 and 244-254 nm in continuous and batch modes, respectively. Continuous flow experiments were conducted at 50% RH, while the RH of batch mode experiments was either 30% or 50%.

A suite of online gas and particle-phase instrumentation was used to monitor concentrations throughout the experiments. Aerosol number and volume concentrations were measured with a scanning mobility particle sizer (SMPS, TSI model 3936), O$_3$ and NO/NO$_2$/NO$_x$ concentrations were monitored with commercial instrumentation (Thermo Environmental Instruments models 49C and 42C, respectively). An Aerodyne high-resolution time-of-flight mass spectrometer (HRToF-AMS) was utilized to measure bulk submicron organic and inorganic aerosol composition.

A high-resolution time-of-flight chemical ionization mass spectrometer (HRToF-CIMS) using iodide adduct ionization was deployed for the detection of both semi- and low-volatility organic compounds (Lee et al., 2014). The HRToF-CIMS was used to monitor the evolving concentrations of both the precursor (IEPOX) and reaction products in both the gas- and particle-phases when coupled to a Filter Inlet for Gases and AEROsols (FIGAERO), from here on referred to as FIGAERO-CIMS. The coupling, optimization, and operation of this combination has been described in detail previously (Lopez-Hilfiker et al., 2014) and is nearly identical to previous operations (D'Ambro et al., 2017a; D'Ambro et al., 2017b). Briefly, the FIGAERO is an

inlet manifold that allows for semi-continuous measurements of both gases and aerosols with
approximately hourly resolution. Aerosol was collected on a 24 mm PTFE filter for 43 minutes
at 2.5 L min$^{-1}$, during which the gases were measured in real time. Following collection,
programmatically heated ultra-high purity (UHP) $N_2$ was passed over the filter, while the
temperature was ramped from ambient to 200 $^{o}$C at a rate of 10 $^{o}$C min$^{-1}$ in order to thermally
desorb compounds from the particle-phase to the gas-phase to be carried into the CIMS for
detection. After the temperature was ramped, it was held at 200 $^{o}$C for 50 minutes to allow
species to desorb and signals to return to background levels. Particle blanks were conducted
approximately every fourth desorption during continuous flow mode or at the beginning and end
of each batch mode experiment by inserting a secondary filter upstream of the primary
FIGAERO filter in order to get a measure of the gas adsorption artifact on the primary filter. Gas
zeros were conducted every 2 minutes by over blowing the CIMS pinhole flow with UHP $N_2$.
The specific coupling to a HRToF-CIMS with iodide ions allows detailed molecular analysis of
hundreds of oxygenated organic compounds via a clustering, fragmentation-free ionization
process.
The FIGAERO-CIMS was also utilized to perform isothermal evaporation experiments as
have been described previously (D'Ambro et al., 2018). In normal operation, the programmatic
thermal desorption is begun immediately after moving the FIGAERO filter under the heating
tube. During isothermal evaporation experiments however, the aerosol is instead exposed for one
hour to a stream of ambient temperature UHP $N_2$ humidified to 50% RH by using a water
bubbler and two mass flow controllers (Figure 1). After the hour exposure, the $N_2$ flow is
reverted to its normal dry state and the programmatic heating proceeds as normal. See Figure 1
for the experimental setup. The RH of 50% during evaporation periods was chosen to match that
of the chamber to keep the phase state of the collected aerosols the same (i.e. so as to not drive
efflorescence). Only the isothermal evaporation portion of the experiment was humidified;
during the thermal desorption the bubbler was isolated using two solenoid valves. Passing excess
humidified $N_2$ over the aerosol and into the instrument resulted in a constant dilution of the
vapor-phase, which allowed for any semi-volatile material to evaporate and also be carried into
the CIMS for detection.

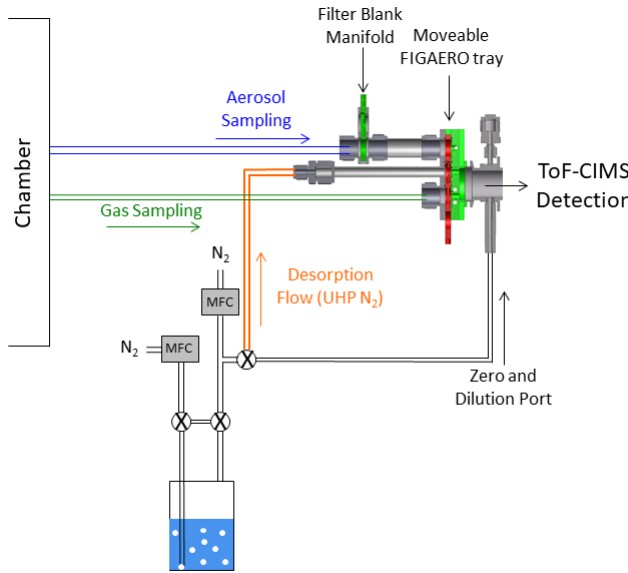


**Figure 1.** Schematic of the FIGAERO isothermal evaporation setup.

**Results & Discussion**

*Chamber-generated IEPOX SOA Composition and Volatility*

Our primary goals with this experiment were to assess whether chamber-generated
IEPOX SOA had a composition and volatility similar to that inferred from field measurements
using the FIGAERO-CIMS of various IEPOX tracers and to test whether IEPOX SOA as a
whole or some of its components were semi-volatile as expected from mechanistic and kinetic
considerations. Overall, the estimated mass yield of OA from IEPOX exposure to aqueous acidic
seed was generally less than unity, of order 0.5 to less than 0.25, somewhat higher than estimates
from Riedel et al [2015]. Mean organic aerosol (OA) mass concentrations generated with IEPOX
and aqueous acidic seed were 2.5 $\mu$g m$^{-3}$ for steady state conditions, 5.5 $\mu$g m$^{-3}$ at the beginning
of the 30% RH batch experiment, and 4.5 $\mu$g m$^{-3}$ at the beginning of the 50% RH batch
experiment. The OA to sulfate ratio was observed to evolve during a batch experiment (e.g.,
decreasing by 20% from the peak), and given the uncertainties associated with potential vapor
wall losses of IEPOX and its reaction products, we refrain from quantitatively interpreting the
SOA yield behaviors in detail.
Regardless of the conditions, the uptake of an authentic IEPOX standard onto acidic
seeds in these experiments results in a rather simple observed particle-phase composition upon
thermal desorption. As shown in Figure 2, a few compositions dominate the average particle-
phase mass spectra, most predominantly $C_5H_{12}O_4$ and $C_5H_{10}O_3$. Species with these compositions
have been repeatedly shown to be major components of IEPOX SOA (Lin et al., 2012; Surratt et
al., 2010; Surratt et al., 2006; Wang et al., 2005), although the relative abundances could change
with time or conditions. In ambient aerosol in the Southeast U.S., the same FIGAERO-CIMS
instrument detected $C_5H_{12}O_4$ and $C_5H_{10}O_3$ in SOA, and these tracers correlated with and
explained ~50% of the IEPOX SOA mass derived from factor analysis of aerosol mass
spectrometer (AMS) data (Lopez-Hilfiker et al., 2016). Our laboratory chamber experiments
starting with an authentic IEPOX standard and acidic seed without photochemical oxidants
therefore support the use of these FIGAERO-CIMS compositions as tracers of IEPOX SOA in
atmospheric particles. As these two compositions are such a large component of the particle-
phase signal (97.5%) measured by FIGAERO-CIMS in chamber generated IEPOX SOA, the
properties of the corresponding SOA are presumably similar to their properties.

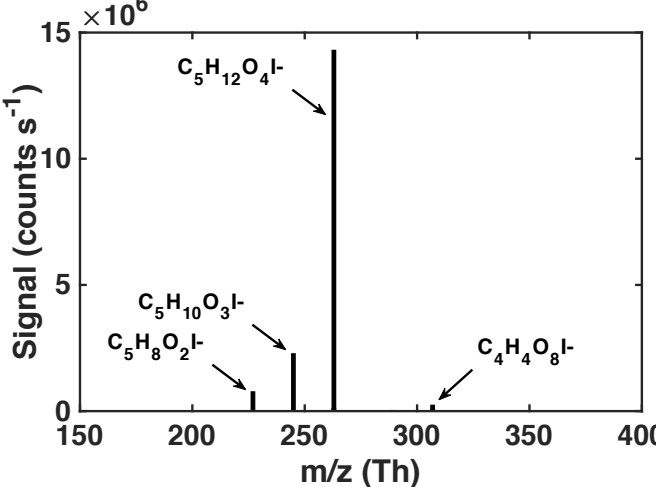


**Figure 2.** Average mass spectrum of IEPOX-derived SOA at a total OA concentration of 5 µg m$^{-3}$.

In previous work, we have related the temperature at which the ion signal for a given molecular composition reaches a maximum during a desorption, known as the $T_{max}$, to the enthalpy of vaporization (or sublimation) and thus effective volatility (Lopez-Hilfiker et al., 2014). Effective volatility refers to the fact that the thermal desorption signal is a convolution of evaporation, thermal decomposition, particle viscosity, and mass transfer of desorbed vapors through the apparatus, and not solely saturation vapor pressure. Notably, the thermogram of $C_5H_{12}O_4$ for chamber generated IEPOX SOA is bimodal (Figure 3, top), i.e. it exhibits two distinct maxima, one occurring at $T_{max}$ = 55 °C (~ 40% of signal), and a second mode at $T_{max}$ = 90 °C. These two maxima indicate SOA components with orders of magnitude different effective volatilities contribute to the desorption of $C_5H_{12}O_4$. For example, using the calibration curve in Lopez-Hilfiker et al. (2014) relating saturation vapor concentration (c*) to $T_{max}$, the two modes correspond to SOA components with an effective c* of 50 and 0.005 µg m$^{-3}$. The lower temperature, higher volatility mode of the $C_5H_{12}O_4$ desorption has a $T_{max}$ consistent with that of an authentic 2-methyltetrol standard, synthesized according to Bondy et al. (2018), deposited on the filter (gray dashed line, Figure 3, top). For comparison, if the structure of $C_5H_{12}O_4$ were assumed to be a 2-methyltetrol, group contribution methods predict the c* to be 34 µg m$^{-3}$ (Compernolle et al., 2011), remarkably similar to what the FIGAERO $T_{max}$-c* relationship predicts for the lower $T_{max}$ mode of the $C_5H_{12}O_4$ thermogram.

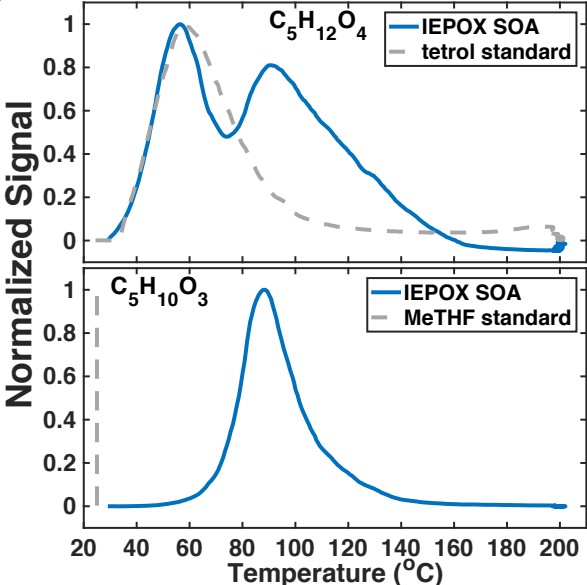

**Figure 3.** Thermal desorption profiles of chamber aerosol (blue) and calibrations with authentic standards (gray, dashed) of the two major particle-phase compounds detected in the chamber: $C_5H_{12}O_4$ (top) and $C_5H_{10}O_3$ (bottom).

The $C_5H_{10}O_3$ thermogram, in contrast, is monomodal, suggesting a single component giving rise to its desorption. However, the $T_{max}$ is much higher than the corresponding authentic *cis*-3-MeTHF-3,4-diol standard, synthesized according to Zhang et al. (2012), (and also that of an alkane triol standard). This standard desorbs completely from the FIGAERO filter in seconds without heating (gray dashed line, Figure 3, bottom). The $T_{max}$ of $C_5H_{10}O_3$ desorbing from IEPOX SOA is 90 °C, the same as the higher $T_{max}$ mode of the $C_5H_{12}O_4$ component, and thus implies a SOA component with an effective c* of at most 0.005 µg m$^{-3}$, indicative of thermal

decomposition during desorption. For comparison, if the structure is assumed to be a $C_5$-alkene triol or 3-MeTHF-3,4-diol, group contribution methods would predict a much larger c* of ~60 or $1.3\times10^5$ μg m$^{-3}$, respectively (Compernolle et al., 2011). For comparison, the group contribution predicted c* of *trans*-β-IEPOX is $1.7\times10^4$ μg m$^{-3}$. All of this evidence indicates that $C_5H_{10}O_3$, as detected in the chamber-generated SOA, is a result of thermal decomposition of a lower volatility species, consistent with recent studies that found the $C_5$-alkene triol was not present as a product of bulk IEPOX reactions (Watanabe et al., 2018) or when the IEPOX SOA was analyzed with novel methods not involving heating (Cui et al., 2018). Taken together, the thermograms of $C_5H_{12}O_4$ and $C_5H_{10}O_3$ therefore indicate a large fraction of the chamber generated IEPOX SOA is composed of very low volatility material (effective c* << 1 μg m$^{-3}$) and several orders of magnitude lower volatility than the compounds that are detected upon thermal desorption.

The thermogram behaviors of chamber generated IEPOX SOA are entirely consistent with those observed for the same tracers measured in ambient aerosol in the Southeast U.S., where $C_5H_{12}O_4$ was also detected with two modes in the thermogram, the respective areas of which varied relative to each other over the course of the measurement campaign (Lopez-Hilfiker et al., 2016). Lopez-Hilfiker et al. (2016) demonstrated from these field observations that the abundance and variability of the lower $T_{max}$ (semi-volatile) mode was consistent with an organic compound having the measured molecular composition and c* of the 2-methyltetrol undergoing equilibrium gas-particle partitioning, while the higher $T_{max}$ mode and that of the $C_5H_{10}O_3$ arose from the decomposition of accretion products.

### *Insights into Volatility via Isothermal Evaporations*

The above considerations of chamber generated IEPOX SOA suggest that a large fraction (the high $T_{max}$ mode) should be relatively stable against evaporation upon dilution of the gas-phase, while the lower $T_{max}$ mode of the $C_5H_{12}O_4$ (likely 2-methyltetrol) component should respond to dilution by evaporating from the particle-phase. To test this hypothesis, we conducted isothermal evaporation experiments using IEPOX SOA generated in the chamber.

Figure 4 shows an example ion signal time series during an isothermal evaporation experiment for $C_5H_{12}O_4$. $C_5H_{12}O_4$ is detected in the gas-phase during particle collection (middle panel, blue shaded area) when chamber air containing IEPOX, acidic aqueous sulfate particles, and IEPOX SOA was being continuously sampled by the FIGAERO-CIMS. The gas-phase sampling included scanning of the dilution ratio, which resulted in varying signals, as well as periodic zeros resulting in occasional significant short-duration drops in signal. We show the last 15-minute portion of the cycle in the gas-phase (blue shaded region) when dilution was held constant, but zeros are still visible as the 2 dips in signal. The chamber was at steady state and the changing gas-phase signal is due to conditioning of the IMR and inlet tubing. During the isothermal evaporation period (yellow shaded area), $C_5H_{12}O_4$ is also detected when a continuous flow of humidified UHP $N_2$ (top panel, 100 = desorption flow on) passes over the particles collected from the chamber on the FIGAERO filter and into the mass spectrometer, consistent with $C_5H_{12}O_4$ evaporation from the collected particles at room temperature (i.e. without heating). Finally, during the temperature-programmed thermal desorption, another pulse of $C_5H_{12}O_4$ was

detected corresponding to components in the remaining SOA that desorbed at elevated
temperature (green shaded area).
The mass concentration of $C_5H_{12}O_4$ measured during a normal temperature-programmed
thermal desorption is compared to that measured during the isothermal evaporation and
subsequent desorption (Figure 4, bottom). Mass closure is achieved to within the experimental
error, driven by variance in normal temperature-programed thermal desorptions due to chamber
conditions and the water vapor effect on CIMS sensitivity (Lee et al., 2014). The observed
behaviors, namely detectable gas-phase concentrations of $C_5H_{12}O_4$ in the chamber and
isothermal evaporation of $C_5H_{12}O_4$ from collected particles indicate that *(i)* $C_5H_{12}O_4$ is produced
from IEPOX reactive uptake, as expected given that the 2-methyltetrol is predicted to be a major
product (Eddingsaas et al., 2010), and *(ii)* a portion of the detected $C_5H_{12}O_4$ behaves as a semi-
volatile organic compound (SVOC), being present in both the gas- and particle-phases, and
evaporating promptly from the particle-phase in response to dilution of the surrounding organic
vapors at room temperature.

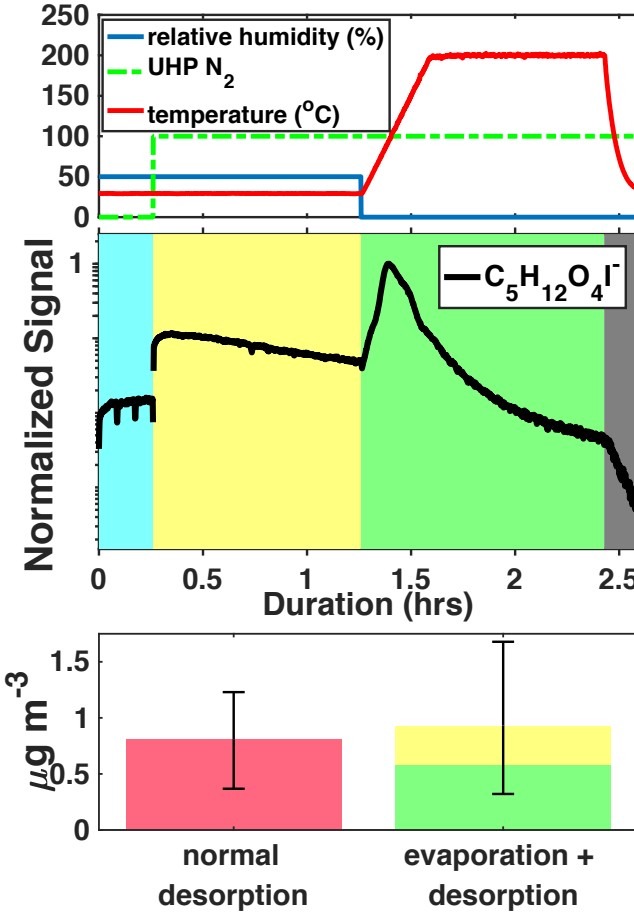


**Figure 4.** Schematic of the isothermal evaporation process. Top: relative humidity (blue), $N_2$
flow (green, 0= off, 100 = on), and temperature (red). Middle: $C_5H_{12}O_4I^-$ during a 1-hour
isothermal evaporation experiment shaded by the phase of the experiments: simultaneous real-
time gas-phase sampling and offline aerosol collection (blue), isothermal evaporation where
compounds are measured as they evaporate off the filter (yellow), temperature-programmed

thermal desorption (green), and cool down of the heating tube (gray). Bottom: mass concentration of $C_5H_{12}O_4$ measured during a normal desorption (pink, left), versus the isothermal evaporation + desorption (yellow and green, right) for the same chamber conditions.

After 1 hour of exposure to UHP $N_2$ at 50% RH, only 16% of the lower $T_{max}$ mode of the $C_5H_{12}O_4$ thermogram remains, suggesting near complete evaporation (Figure 5, top). The observed rate of decay (84% hr$^{-1}$) upon dilution can be related to an effective c*, or, if the structure is 2-methyltetrol it is likely to partition into the aqueous-phase, making an effective Henry's law constant more appropriate. Assuming no particle-phase diffusion limitations but accounting for FIGAERO mass transfer limitations (Schobesberger et al., 2018), we predict a c* of 5-15 μg m$^{-3}$ for the portion of the $C_5H_{12}O_4$ thermogram that evaporates. Utilizing COSMOtherm (2018) with the BP_TZVPD_FINE_18 parameterization as described previously (Kurtén et al., 2018), a Henry's law constant of $4.9\times10^8 - 1.1\times10^{10}$ M atm$^{-1}$ is predicted if all conformers are used ($4.9\times10^8$ M atm$^{-1}$) or if the number of internal H-bonds is minimized ($1.1\times10^{10}$ M atm$^{-1}$), which compares well to the value calculated from the observed decay of $C_5H_{12}O_4$ during the evaporation ($1.8\times10^8$ M atm$^{-1}$). These estimates do not include the likelihood of a salting-out effect expected for 2-methyltetrol (Waxman et al., 2015), which would further lower the Henry's Law constant. Whether Raoult's or Henry's law is the appropriate framework for interpretation depends on whether IEPOX reactive uptake results in a phase-separated organic medium, for example an organic coating, or a homogeneous aqueous solution, and the competitive partitioning of the $C_5H_{12}O_4$ species between two such regimes. Regardless, as we show below, the semi-volatile nature of low $T_{max}$ portion of $C_5H_{12}O_4$, assumed to be the 2-methyltetrol, a major product of IEPOX reactive uptake, will cause it to partition strongly to the gas-phase under typical atmospheric conditions outside of cloud.

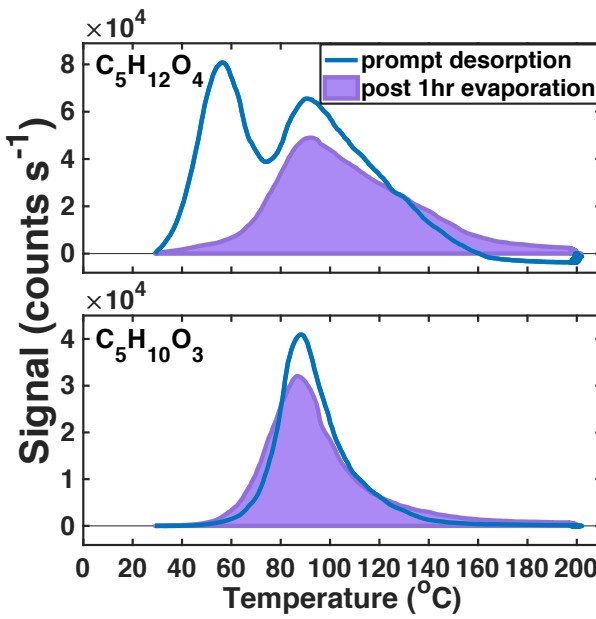

**Figure 5.** Thermograms obtained from prompt desorption of the aerosol (blue) and after one hour of evaporation (lavender, shaded) of the two major particle-phase compounds detected in the chamber: $C_5H_{12}O_4$ (top) and $C_5H_{10}O_3$ (bottom).

The second, higher $T_{max}$ mode of the $C_5H_{12}O_4$ thermogram and the single Gaussian-like thermogram of $C_5H_{10}O_3$ (Figure 5, bottom) do not change significantly after 1-hour of exposure to UHP $N_2$. In the case of the $C_5H_{10}O_3$ thermogram, nearly 95% of signal remains after the 1-hour evaporation period. In both cases, there is a slight broadening of the thermogram and a measurable increase in material desorbing at the highest temperatures (120+ $^o$C). Previous isothermal evaporation experiments with the FIGAERO have shown that, for α-pinene ozonolysis SOA, the physical age of the aerosol had a controlling role in the volatility of the bulk SOA and individual desorbing compounds (D'Ambro et al., 2018), consistent with the additional hour of non-oxidative aging in this system resulting in an increase in lower-volatility material. As to the widening of the thermograms, Schobesberger et al. (2018) showed that a shallowing of the low temperature side of the thermogram and a broadening of the higher temperature tail correspond to thermal decomposition from a larger suite of bonds with different dissociation energies, consistent with continued formation of a variety of accretion products during the isothermal evaporation period.

The above evidence supports the previous assertions of Lopez-Hilfiker et al. (2016) that the lower $T_{max}$ mode of the $C_5H_{12}O_4$ thermogram corresponds to a semi-volatile component, very likely the 2-methyltetrol, and further support the conclusion that IEPOX SOA in ambient aerosol is very to extremely low volatility (Lopez-Hilfiker et al., 2016; Hu et al., 2016; Riva et al., 2019). The isothermal evaporation experiments presented above provide an explanation as to why the ambient SOA contained such a relatively small fraction of the low $T_{max}$ (semi-volatile) 2-methyltetrol component in that it likely had evaporated to maintain gas-particle equilibrium. Furthermore, the high $T_{max}$ tracers do not decay in abundance during the evaporation experiments, but rather slightly increase with time at the highest temperatures (>120 $^o$C), indicating ongoing accretion chemistry leading to lower volatility components.

*Effect of RH and Acidity on IEPOX SOA Characteristics: Mechanistic Insights*

We performed two time-dependent "batch mode" chamber experiments using IEPOX and acidic aqueous seed particles, one at 30% and the other at 50% RH. By operating in batch mode as opposed to continuous flow mode, we are able to temporally resolve the formation of SOA. By varying the RH, we simultaneously varied the liquid water content relative to sulfate, and therefore also acidity. Three sequential thermal desorptions of $C_5H_{12}O_4$ obtained over the course of experiments (~10 hrs total) at 30% (left) and 50% (right) RH are shown in Figure 6, top. At 30% RH, the lower $T_{max}$ (higher-volatility) mode grows in rapidly and is clearly visible in the first desorption (black line). The first desorption occurred after 43 minutes of particle collection, which began immediately after the seed was injected into the chamber and IEPOX uptake was initiated, resulting in collected aerosols having a variety of ages and thus the median age of 22 min is assumed. However, this mode then does not grow significantly larger as the experiment progresses. During the 2[nd] and 3[rd] desorptions, 2 hrs 22 min and 4 hrs 22 min respectively, after the initial exposure, the higher $T_{max}$ (lower-volatility) mode is visible and dominates the thermogram.

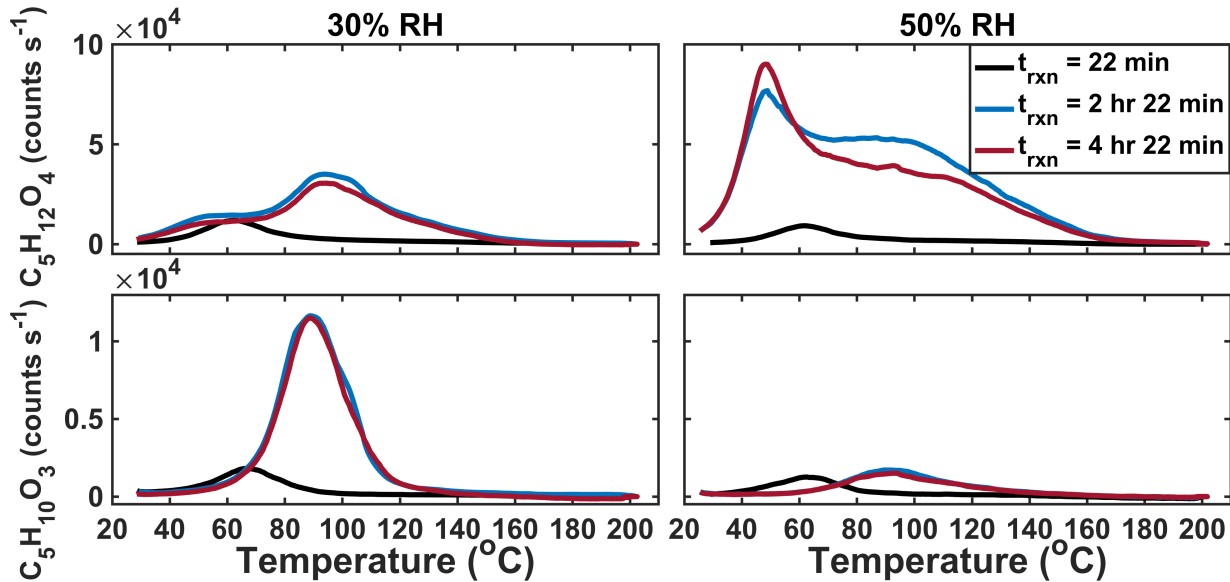

**Figure 6.** Sequential desorptions of $C_5H_{12}O_4$ (top) and $C_5H_{10}O_3$ (bottom) during batch mode experiments at 30% RH (right) and 50% RH (left).

In the 50% RH batch experiment, the lower $T_{max}$ (higher-volatility) mode of the $C_5H_{12}O_4$ thermogram also dominates in the first desorption, but in contrast to the experiment at 30% RH, this lower $T_{max}$ mode continues to grow and is the dominant portion of the thermogram for all desorptions. While the higher $T_{max}$ mode is observed after the 1st desorption, it is much broader and has an ambiguous peak, unlike at 30% RH. In the 30% and 50% RH experiments, both modes of the thermogram are observed by the 2nd desorption 2.5 hours after IEPOX uptake starts, but the relative abundance and shape of the two modes differ with RH. The thermogram shape also changes as a function of time since IEPOX injection in the steady state experiments, although due to the range in aerosol ages present within the chamber at a given time during steady-state experiments, it is more straightforward to define this feature as a function of time in batch mode measurements. The corresponding thermograms of $C_5H_{10}O_3$ in each of the two batch mode experiments are shown in Figure 6, bottom. Most obvious is that the amount of $C_5H_{10}O_3$ desorbing, relative to the $C_5H_{12}O_4$, is highest in the 30% RH experiment, when sulfate and hydronium ion concentrations are highest. Further work could be done to understand the evolution of IEPOX SOA components as a function of time, but a fairly stable set of products and volatility are reached within a few hours.

Along with the shape, the $T_{max}$ of the $C_5H_{10}O_3$ and each mode of the $C_5H_{12}O_4$ vary slightly in time, the two of which are likely related. It has been shown previously that when the IEPOX-derived organosulfate ($C_5H_{12}SO_7$) is deposited on and desorbed from the FIGAERO filter, it decomposes into both $C_5H_{12}O_4$ and $C_5H_{10}O_3$. The corresponding $T_{max}$ of both are co-located and highly dependent on acidity, with higher acidity leading to lower $T_{max}$'s (Lopez-Hilfiker et al., 2016). This dependence on the inorganic aerosol components, present in much larger excess in these experiments than our previous FIGAERO experiments, could be the cause of the shifts of the lower $T_{max}$ modes. Alternatively, the shift could be due to the increasing complexity of the SOA as it evolves in time leading to different interactions between particle components which affects volatility.

From these observations we can draw two conclusions regarding the mechanisms that give rise to the dominant components of IEPOX SOA, illustrated in Figure 7 which shows

hypothetical reaction pathways and oligomers that could explain the observed time-evolution of
detected products. First, the low $T_{max}$, semi-volatile $C_5H_{12}O_4$ exists in the aerosol from the first
desorption and thus is likely formed promptly from IEPOX uptake, consistent with the formation
of 2-methyltetrol via nucleophilic attack by water of the protonated epoxide ring (see Figure 7).
That this semi-volatile mode is more prominent in the higher RH experiment, i.e. higher liquid
water content and therefore higher $H_2O$-nucleophile content relative to sulfate, further supports
this interpretation. Additionally, the higher liquid water content supports a greater amount of 2-
methyltetrol remaining partitioned in the aerosol via Henry's Law, consistent with offline filter
analysis (Riva et al., 2016). Second, the higher $T_{max}$ (lower volatility) modes are mostly
produced more slowly over time, indicating a second or higher generation product of IEPOX
uptake, as these modes are mainly observed 2.5 hours after IEPOX uptake has largely ended.
Thus, if the higher $T_{max}$ modes are from the thermal decomposition of an organosulfate product,
as suggested by Lopez-Hilfiker et al. (2016) and as Cui et al. (2018) demonstrate, our
experiments suggest it is unlikely to form solely from nucleophilic addition of (bi-)sulfate to
protonated IEPOX, as that reaction should occur concurrently with 2-methyltetrol formation.

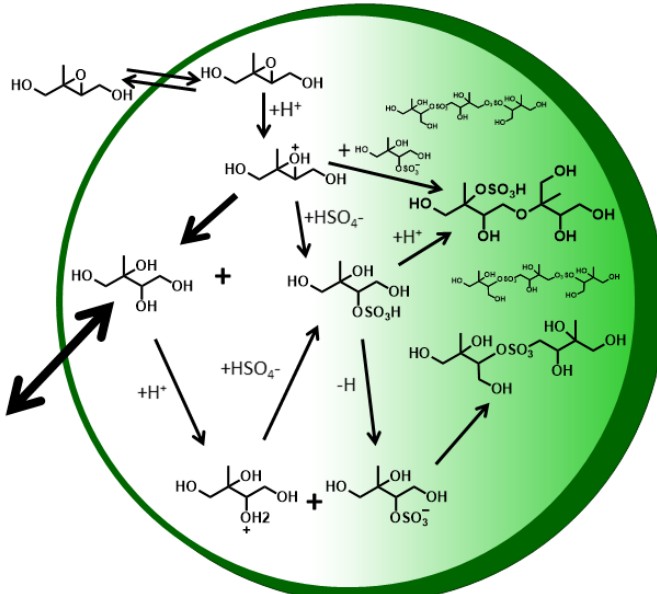

**Figure 7.** Hypothetical schematic of IEPOX reactive uptake and particle-phase processes. White
region denotes semi volatile species that actively partition between the gas- and particle-phases,
light green denotes species that are of lower volatility, and dark green outline denotes a coating.
453    Another possible mechanism of organosulfate formation, as well as sulfate ester
oligomers, is via $S_N2$ reactions where one of the 2-methyltetrol –OH groups is protonated to
make $H_2O$ the leaving group while bisulfate, sulfate, or an organosulfate is the substituting group
(Figure 7). At 30% RH, the particle acidity is higher due to less dilution of the sulfate, which
would result in higher organosulfate concentrations and acid catalyzed accretion chemistry (Jang
et al., 2002), consistent with the observation of a more prominent higher $T_{max}$ (lower volatility)
mode compared to the 50% RH experiment. The broader higher $T_{max}$ mode at 50% RH indicates
that there is likely an array of compounds breaking apart to give rise to this specific composition,
consistent with a greater variety of oligomerization reactions occurring due to the dilution of
sulfate and higher 2-methyltetrol concentrations. Previous work has identified several non-sulfur
containing polyol species, both monomers and oligomers, in IEPOX SOA (Surratt et al., 2010;
Lin et al., 2012; Lin et al., 2014).
**Summary & Atmospheric Implications**
To place our findings into context, we present results from a simple conceptual model
simulating IEPOX (initially ~2 ppb/ 10 μg m$^{-3}$) reactive uptake to form the corresponding 2-
methyltetrol and organosulfate at yields of 90 and 10%, respectively, with an uptake coefficient
of 0.05 based on Gaston et al. (2014), and an atmospherically relevant total surface area (2.5×10$^{-}$
$^{6}$ cm$^2$ cm$^{-3}$) and volume (1.6×10$^{-11}$ cm$^3$ cm$^{-3}$). We also include a loss of gas-phase species (2-
methyltetrol and IEPOX) due to reaction with OH at 1.8×10$^{-11}$ cm$^3$ molec$^{-1}$ s$^{-1}$ (Atkinson, 1987),
consistent with previous studies for the IEPOX + OH rate constant (Bates et al., 2014; Jacobs et
al., 2013). The processes in the model are simplified from the reaction scheme discussed above,
i.e. it does not include particle-phase processes, but its purpose is to capture the salient factors
that control the reactive uptake and partitioning. The chosen branching between 2-methyltetrol
and organosulfate yields from IEPOX reactive uptake to aerosol is somewhat arbitrary and only
to illustrate the behavior of the system. The Henry's Law constant found via COSMOtherm for
the 2-methyltetrol is used to simulate gas-particle partitioning of the 2-methyltetrol, while the
organosulfate is a proxy for all low volatility products, including the promptly formed
organosulfates, sulfate esters from further accretion, and polyol oligomers. Vapor wall loss is not
considered in the model, which might be resulting in more tetrol evaporating from the particles
in our measurements than would occur in the atmosphere. However, operating in continuous
flow mode helps to mitigate these issues, and in batch mode we do not observe a significant loss
of the low-$T_{max}$/semi-volatile mode. In the atmosphere, photochemical losses of the gas-phase
tetrol and the smaller aqueous volume of the aerosol would lead to partitioning of the tetrol out
of the aerosol, as illustrated by the model in Figure 8 which does include these atmospheric
processes. Thus, while vapor wall loss in the chamber potentially leads to lower particle-phase
tetrol, the chamber experiments neglect oxidation of gas-phase tetrol which would have a similar
effect in the atmosphere.
The simulated loss of IEPOX and formation of IEPOX SOA are shown in Figure 8.
IEPOX is almost completely consumed after 1 hour of reaction, corresponding to rapid formation
of SOA. The composition of the aerosol changes significantly as a function of time. Initially, the
SOA is composed primarily of 2-methyltetrol. However, despite the relatively high Henry's law
constant, much of the 2-methyltetrol evaporates into the gas-phase to maintain equilibrium with
the gas-phase 2-methyltetrol which is subjected to loss by gas-phase bimolecular reactions with
the hydroxyl radical (OH). This behavior supports our findings herein that on a typical aerosol
lifetime, the dominant IEPOX reactive uptake product, 2-methyltetrol, will be a small component
of IEPOX SOA, and organosulfates and other low volatility material, including oligomers of 2-
methyltetrol, will dominate, albeit at a smaller overall SOA yield (Figure 8) per IEPOX reacting
on aerosol.

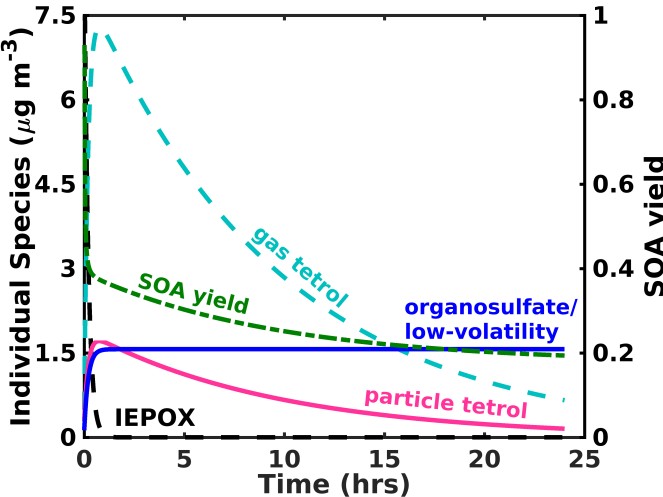

**Figure 8.** Model results for the major gas- and particle-phase species of IEPOX reactive uptake for typical atmospheric aerosol and typical IEPOX mixing ratios.

Fundamental chamber studies of IEPOX reactive uptake to aqueous acidic seed were performed and we find that the resulting molecular composition and volatility of the formed SOA suggests that the vast majority of IEPOX SOA in the atmosphere is of very low volatility, in the form of organosulfates and polyol oligomers, which is supported by the recent findings of Riva et al. (2019). We show that the major product expected from IEPOX reactive uptake in acidic aqueous solutions, $C_5H_{12}O_4$ (most probably 2-methyltetrol), is semi-volatile and likely will partition out of the aerosol and thus contribute relatively little to IEPOX SOA mass under atmospheric conditions. We further confirm that the observed properties of $C_5H_{10}O_3$ are not consistent with the structure of $C_5$-alkene triols and/or 3-MeTHF-3,4-diols, and thus these structures cannot be components of IEPOX SOA but are likely artifacts of thermal decomposition during analytical workup. While our results are specific to the FIGAERO, we predict that the issue is more general, affecting other methods, as indicated for example in Cui et al. (2018). A direct intercomparison is required to definitively determine whether all instruments (those reporting $C_5H_{12}O_4$, $C_5H_{10}O_3$, 2-methyltetrols, $C_5$-alkene triols, and/or 3-MeTHF-3,4-diols) are in fact measuring the same species and that prior estimates of IEPOX SOA have not been overestimated due to "double counting" carbon in these tracers which might be derived from organosulfates and oligomers measured separately.

The evidence presented herein, as well as in independent experiments (Cui et al., 2018), indicates that the $C_5H_{10}O_3$, regardless of structure, as well as a significant portion of the $C_5H_{12}O_4$, are not actual components of the SOA but rather derived from other related components during the analysis. Therefore, we do not recommend that these species be included as products in mechanistic models of IEPOX SOA formation and evolution. Finally, we provide evidence that a portion of the low volatility IEPOX SOA is composed of oligomers formed in part from slower particle-phase accretion chemistry, likely involving the first generation organosulfates and possibly also the 2-methyltetrol, though we can't distinguish between these possible combinations. The distribution of products and formation timescales depend upon aerosol water, sulfate, and hydronium ion activities, and thus ultimately on ambient RH and particle alkalinity sources. However, the low volatility of the majority of IEPOX SOA will make it less susceptible to subsequent changes in RH or dilution.

These findings help to explain properties of IEPOX SOA observed in the field and to resolve inconsistencies in the descriptions of IEPOX SOA formation from gas-phase IEPOX reactive uptake. The AMS IEPOX related PMF factor has been shown to correlate with sulfate (Xu et al., 2015; Hu et al., 2015). If the majority of IEPOX SOA is in the form of low volatility components, such as organosulfates and oligomers thereof as our observations indicate, then a correlation of IEPOX SOA with sulfate, and not water content or acidity, is expected given the direct coupling to inorganic sulfate and similar lifetimes against removal. Moreover, the low SOA yield per reactive uptake of IEPOX on aqueous acidic seed can be explained as $C_5H_{12}O_4$, the dominant product and likely 2-methyltetrol, being semi-volatile and largely partitioning out of the particle-phase especially in a system with gas-phase oxidation or in chambers with vapor-wall loss. Measured reactive uptake probabilities of IEPOX ($\gamma_{IEPOX}$) on aqueous acidic seed are an order of magnitude or more higher than often used in models (Gaston et al., 2014; Eddingsaas et al., 2010; Pye et al., 2013; Marais et al., 2016). Partly this discrepancy reflects a lower SOA yield per IEPOX lost than accounted for in models, for which we provide an explanation, and also a role for organic coatings that likely exist in the atmosphere which slow reactive uptake relative to pure aqueous acidic seed (Gaston et al., 2014; Zhang et al., 2018).

These aspects likely have some cancelation of errors but could lead to errors in models of IEPOX abundance and the IEPOX SOA production rate, and thus their corresponding spatial variability. For example, in regions with relatively fresh, uncoated aqueous acidic particles, models using a low reaction probability would underestimate the IEPOX loss rate from the gas-phase, sustaining IEPOX SOA formation over a larger area than in reality. A model that ignored the role of organic coatings and 2-methyltetrol evaporation would potentially overestimate IEPOX SOA formation on regional scales. Models that treat products of IEPOX reactive uptake in a volatility basis set would need to utilize the volatility inferred from the thermograms herein, or thermal denuder measurements, rather than that inferred from commonly reported IEPOX SOA tracers that utilize high-temperature analytical methods. Finally, without accounting for slower accretion chemistry involving organosulfates and polyols, IEPOX SOA predicted from a treatment of first-generation products alone could be underestimated.

**Author Contributions**

E.L.D. analyzed FIGAERO data; E.L.D., J.A.T., and S.S. conducted modeling; and E.L.D. and J.A.T. wrote the manuscript. J.E.S. and J.A.T designed the chamber experiments. N.H., V-T. S., G.H., and T.K made COSMOtherm predictions. All other coauthors participated in data collection, experiment operations, and manuscript discussions.

**Acknowledgements**

This work was supported by the U.S. Department of Energy ASR grants DE-SC0011791 and DE-SC0018221. E.L.D was supported by the National Science Foundation Graduate Research Fellowship under Grant No. DGE-1256082. PNNL authors were supported by the U.S. Department of Energy, Office of Biological and Environmental Research, as part of the ASR program. Pacific Northwest National Laboratory is operated for DOE by Battelle Memorial Institute under contract DE-AC05-76RL01830. UC Davis authors were supported by NSF grant ATM-1151062. We thank A. Gold and Z. Zhang for synthesizing the *trans*-β-IEPOX, 2-methyltetrol, and 3-MeTHG-3,4-diol standards used in this work.

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

# Figure Captions

**Figure 1.** Schematic of the FIGAERO isothermal evaporation setup.

**Figure 2.** Average mass spectrum of IEPOX-derived SOA at a total OA concentration of 5 μg m$^-$$^3$.

**Figure 3.** Thermal desorption profiles of chamber aerosol (blue) and calibrations with authentic standards (gray, dashed) of the two major particle-phase compounds detected in the chamber: $C_5H_{12}O_4$ (top) and $C_5H_{10}O_3$ (bottom).

**Figure 4.** Schematic of the isothermal evaporation process. Top: relative humidity (blue), $N_2$ flow (green, 0= off, 100 = on), and temperature (red). Middle: $C_5H_{12}O_4I$- during a 1-hour isothermal evaporation experiment shaded by the phase of the experiments: simultaneous real-time gas-phase sampling and offline aerosol collection (blue), isothermal evaporation where compounds are measured as they evaporate off the filter (yellow), temperature-programmed thermal desorption (green), and cool down of the heating tube (gray). Bottom: mass concentration of $C_5H_{12}O_4$ measured during a normal desorption (pink, left), versus the isothermal evaporation + desorption (yellow and green, right) for the same chamber conditions.

**Figure 5.** Thermograms obtained from prompt desorption of the aerosol (blue) and after one hour of evaporation (lavender, shaded) of the two major particle-phase compounds detected in the chamber: $C_5H_{12}O_4$ (top) and $C_5H_{10}O_3$ (bottom).

**Figure 6.** Sequential desorptions of $C_5H_{12}O_4$ (top) and $C_5H_{10}O_3$ (bottom) during batch mode experiments at 30% RH (right) and 50% RH (left).

**Figure 7.** Hypothetical schematic of IEPOX reactive uptake and particle-phase processes. White region denotes semi volatile species that actively partition between the gas- and particle-phases, light green denotes species that are of lower volatility, and dark green outline denotes a coating.

**Figure 8.** Model results for the major gas- and particle-phase species of IEPOX reactive uptake for typical atmospheric aerosol and typical IEPOX mixing ratios.