# Peer review of "Chamber-based insights into the factors controlling IEPOX SOA yield,"

_Atmospheric Chemistry and Physics, 2019_

## Short Comment (SC1) · 25 Apr 2019

Comments prepared by Lindsay Yee, Gabriel Isaacman-VanWertz, and Allen Goldstein

In light of the uncertainty surrounding the molecular structure(s) that actually lead to observations of C5-alkene triols in chromatography based techniques and C5H10O3 measurements in CIMS, we think that there are areas of your manuscript that should be adjusted per below.

I. Citation of literature:
   A. Line 87: Reference to Isaacman-VanWertz et al., ES&T 2016 would be appropriate here
   B. Section starting line 289: Why not also compare results to field measurements of volatility as in Hu et al, ACP, 2016, DOI: 10.5194/acp-16-11563-2016?
   C. Consider providing additional chemical understanding of proposed compounds in Figure 7 and their expected behavior under programmed thermal desorption and hydrolysis affecting oligomer/accretion products recovery as in Claflin and Ziemann et al., AS&T, 2019, DOI: 10.1080/02786826.2019.1576853. Are your isothermal evaporation results with humidified air providing opportunity for hydrolysis? While lines 52-56 state that your methods confirm that these products are artifacts of thermal decomposition/hydrolysis, hydrolysis is not discussed explicitly elsewhere in the manuscript as it applies to your measurements.
   D. Lines 224-227: Reword to be more accurate. Lopez-Hilfiker et al., 2016, pg. 2204 states that a sum of tracers in CIMS measurement that are highly correlated with C5H12O4 and C5H10O3 explain 50% of IEPOX-SOA factor mass, not just C5H12O4 and C5H10O3.
   E. Line 455: Do you mean Lin et al., ES&T 2014, DOI: 10.1021/es503142b, instead of Lin et al., ES&T 2012, or in addition?

II. Be more precise in language throughout the manuscript relating instrument's observation of a chemical formula (which even stated in line 47 that C5H12O4 is "presumably" 2-methyltetrols) yet in other areas claim unwarranted certainty in the chemical structure of C5H12O4 and C5H10O3. This also needs to be adjusted regarding relation to findings under chamber conditions and then related to atmosphere. Language should be adjusted throughout; some examples specifically listed below:

A. Lines 52-56: "We thus confirm, using controlled laboratory studies, recent analyses of ambient SOA measurements showing that IEPOX SOA is of very low volatility and commonly measured IEPOX SOA tracers, such as methyltetrols and C5-alkene triols, result predominantly from artifacts of measurement techniques associated with thermal decomposition and/or hydrolysis." As it is not verified if the C5-alkene triol structure is what gives C5H10O3 in CIMS (standards don't exist yet, though semi-reasonable to assume from a chamber experiment), this statement is not supported enough. Further, Surratt's HILIC method (Cui et al., Environ. Sci. Process. Impacts, 2018) which does measure some higher order organosulfates and oligomers still does not account for all C5-alkene triols as stemming from decompositon (<50%), so is it fair to say predominant? Maybe just predominant for chamber conditions tested here?

B. Lines 220-224: The chemical formulae observed by FIGAERO CIMS are consistent with those from actual chemical species observed by other techniques. Line 221 should be adjusted to something more like, "These chemical formulae are consistent with those of chemical species (i.e. 2-methyltetrols and C5-alkene triols, respectively) that have been repeatedly shown to be major components of IEPOX SOA…"

C. While lines 270-278 are precisely worded to state that the interpretation of C5H10O3 as observed in chamber SOA derives primarily from thermal degradation, this does not warrant broad generalization to the atmosphere as in lines 52-56, lines 494-497. Considering there are other potential chemical conditions under which C5-alkene triols may be observed, but not tested here there should be room left for these possibilities, even if less abundant in the atmosphere including:

    i. IEPOX uptake on non-acidified seed/dry particles:

        1. Nguyen et al., ACP, 2014, DOI: 10.5194/acp-14-3497-2014

        2. Riva et al., ES&T 2016, DOI: 10.1021/acs.est.5b06050 (Fig. 2 shows C5-alkene triols observed under dry conditions and wet conditions)

        3. Lin et al., ES&T 2014, DOI: 10.1021/es503142b (Fig. S2 shows C5-alkene triols observations under wet_netural and dry_acidic conditions)

        4. D'Ambro et al., ACP, 2017, DOI: 10.5194/acp-17-159-2017 (Fig. 4 shows C5H10O3 observation)

    ii. Non-IEPOX pathways:

        1. Liu et al., ES&T 2016, DOI: 10.1021/acs.est.6b01872, as C5H10O3 is also observed here reported in Figure 3

        2. Riva et al., ES&T 2016, DOI: 10.1021/acs.est.6b02511, as C5-alkene triols are observed as reported in Figure 1

    iii. isoprene ozonolysis leading to structures proposed as sulfate esters of C5-alkene triols as in Riva et al., Atmos. Env. 2016, DOI: 10.1016/j.atmosenv.2015.06.027)

III. The connection of FIGAERO-CIMS measurement of C5H10O3 being equivalent to the same C5-alkene triols signal reported from GC-MS is not clear unless an intercomparison was done previously.  If done, then please cite the reference:

    A. Lines 494-497: Claim of confirmation that C5-alkene triols and/or 3-MeTHF-3,4-diols are all artifacts of thermal decomposition is too strong given that it has not been established if what the FIGAERO CIMS measures as these compounds (C5H10O3) is actually the same as what other techniques (GC/MS) would assign to be C5-alkene triols and/or 3-MeTHF-3,4-diols.  Where is there direct comparison of FIGAERO-CIMS measured C5H10O3 and other techniques measured C5-alkene triols and/or 3-MeTHF-3,4-diols to prove that the compound/s is/are the same?

    B. Further, this means the conclusive statement in lines 522-523 is not supported to broadly overgeneralize that GC/MS methods are incorrect.  As there are a variety of ways in which GC/MS methods are employed to measure isoprene-derived tracers as well as how the samples are handled and treated before GC/MS analysis, it does not seem appropriate to mention this here without performing proper intercomparisons or going into more specifics as to why all GC/MS methods would lead to thermal decomposition of these products.   Thus, statements about specific tracers being "artifacts" or somehow not being useful measurements are misleading and unnecessary. It seems more productive to focus on what the community can learn from the different methods, and what the different tracers teach us, rather than denigrating a particular measurement approach because it sees a combination of products that may include pieces that decompose from larger molecules.  For example, the UPLC/HILIC/GC/MS methods provide more specific information (more highly speciated EI mass spectrum of individual tracers that elute from chromatography columns) than the direct CIMS (sum of isomers as a function of desorption temperature) or AMS (EI mass spectrum of the total), but using all the observations we can learn more (not less) about the chemical composition, sources, and transformation processes in organic aerosols.

IV. For some of the structures (organosulfates/polymers) proposed to lead to C5H10O3 via decomposition, can you estimate their C*/Tmax and do they make sense with the observed thermograms?

---

## Referee Comment (RC1) · Anonymous Referee #1 · 3 May 2019

General comments

This manuscript describes laboratory experiments using the Filter Inlet for Gases and Aerosols/Chemical Ionization Mass Spectrometry (FIGAERO-CIMS) technique that aim to investigate the nature of the components of IEPOX-derived secondary organic aerosol (IEPOX-SOA). Specifically, the work addresses the inconsistency between GC-MS approaches that have identified semi-volatile molecular components and volatility measurements that have indicated that the bulk of IEPOX-SOA must be made up of much lower volatility molecular components. The main claim is that a desorption signal that corresponds to $C_5H_{12}O_4$ has two maxima, one that arises from a semi-volatile

source (presumably from 2-methyl tetrols directly evaporating) and one that arises from the thermal decomposition of a low volatility source and a desorption signal that corresponds to C5H10O3, which also arises from the thermal decomposition of a low volatility source. Because knowledge of the molecular composition of IEPOX-SOA is critical to the development of accurate SOA mechanistic models, this work will be of great interest to readers of Atmospheric Chemistry and Physics. However, I believe that a number of uncertainties remain in the interpretation of both the present work and past studies, and that a revised manuscript should more directly address these issues.

Specific comments

Because of a lack of authentic standards, there continues to be no proof whatsoever that either the GC/MS signals previously attributed to C5-alkene triols or the present CIMS signals attributed to C5-alkene triols actually correspond to these species. Indeed, Watanabe et al. 2018 showed that these species are among the least thermodynamically favored among a variety of possible C5H10O3 isomers. I suggest that the manuscript be revised to simply refer to a C5H10O3 thermal decomposition product and refrain from associating this product with any particular molecular form.

Similarly, while I find the argument fairly convincing that the semi-volatile C5H12O4 component is probably the 2-methyl tetrols themselves, I don't think the low volatility thermal decomposition product can be assumed to be the 2-methyl tetrols.

The proposed oligomerization mechanism given in Figure 7 would benefit from more detailed discussion. Rather than a more obvious mechanism directly involving IEPOX, the authors are proposing two types of reactions: 1) acid-catalyzed etherification reactions of organosulfates to form ether-linked oligomers and 2) acid-catalyzed sulfate esterification reactions to form sulfate-linked oligomers. The authors don't provide any literature precedents for these types of reactions. Therefore, I think the authors should provide a rationale for these somewhat unusual reaction types.

[Figure]

Along the same lines, the overall interpretation would benefit from estimates of the thermal desorption behavior of the proposed oligomer components. Did the authors suggest two isomers of a monosulfate dimer as their major proposed molecular species because C10H22SO10 is expected to have roughly the observed thermal desorption behavior?

Technical comments

Line 87: There should be a reference to Hu et al. 2016 ACP 16, 11563-11580 here.

Line 128: There is something wrong with the grammar in this sentence. Please revise.

Line 464: I'm not sure why there is a reference to Atkinson, 1987 here. There are two measurements of IEPOX + OH rate constants: Bates et al. 2014 and Jacobs et al. 2013, ES&T 47, 12868-12876, which differ by a factor of two. The choice of rate constant should be explicitly discussed.

---

## Referee Comment (RC2) · Anonymous Referee #2 · 3 May 2019

Overview:

This study sheds new light on the question of volatility and chemical composition of secondary organic aerosol derived from IEPOX, a ubiquitous biogenic aerosol component. IEPOX is known to react in aqueous aerosol to form commonly observed products such as methyltetrols and organosulfates, and compounds with the molecular formula $C_5H_{10}O_3$ (C5 alkene triols, methyl-tetrahydrofuran-diols, or both) have also been observed in IEPOX SOA. The total reactivity of IEPOX in the aerosol phase and its uptake are known to be highly dependent on aerosol acidity, sulfate content, organic coating, and other parameters, but a detailed chemical understanding of the

composition of the resulting organic aerosol is still lacking. Most observations of ambient IEPOX-derived SOA suggest that the majority of the SOA is of lower volatility than the individual species described above, suggesting a greater presence of oligomers and organosulfates and relatively small contributions from methyltetrols, alkene triols, or MeTHF-diols. This work bridges that apparent gap by showing that observed alkene triols or MeTHF-diols are artifacts formed during the thermal decomposition of lower-volatility material, as is a portion of the signal observed at the same mass as methyltetrols, while another portion that likely arises from the methyltetrols themselves is volatile and undergoes evaporation from the particle phase on the timescale of 1 hour.

The authors reach these conclusions by observing the uptake of trans-b-IEPOX onto acidified ammonium bisulfate seed aerosol in a series of controlled chamber experiments. They evaluate the evolution of particle chemical composition and component volatility using a high-resolution time-of-flight I- chemical ionization mass spectrometer outfitted with a filter inlet for gases and aerosols, which enables the separation of isobaric compounds by thermal desorption. Thermograms taken at varying humidities and timepoints during the experiments support the assertion that methyltetrols form quickly via direct hydrolysis following IEPOX uptake, and are then both transformed to lower-volatility species and evaporated into the gas phase, where they can be lost to walls or further photooxidation. The remaining IEPOX-derived SOA is therefore of very low volatility, likely including oligomers of the tetrols and organosulfates. Finally, the authors use a simple box model to illustrate how these results might play out under ambient conditions, and show that models using either aqueous uptake or volatility basis set schemes should be modified to account for the high uptake probability of IEPOX followed by revolatilization of methyltetrols.

While the description of the experiments performed herein and the conclusions drawn from thermal desorption measurements is straightforward and represents a valuable contribution to our understanding of the volatility and chemical composition of IEPOX-derived SOA, the extrapolation to ambient conditions remains tenuous and deserves

further attention. Additional comparison to ambient results (e.g. Hu et al 2016) would help convince the reader that the results from these chamber studies are borne out in the atmosphere and are relevant to processes that occur in isoprene- and sulfate-rich settings. The brief comparison on lines 221-232 is not sufficient, and if anything brings up more questions than it answers as to the validity of comparing field to laboratory results. Does Lopez-Hilfiker et al. suggest that a further 50% of IEPOX SOA observed in ambient consists of tracers that don't even correlate with those identified here, in addition to the fraction that correlates with them but isn't explicitly C5H12O4 or C5H10O3? What are those other tracers and what could explain their absence in these experiments? Ambient particles likely contain much more variability in organic compounds in the condensed phase with which tetrols and other IEPOX-derived species might oligomerize or otherwise interact. How would that affect the conclusions here and our ability to put simple parameterizations into models? Is there any evidence that products of such cross-reactions result in similar observed compounds upon thermal decomposition?

Technical comments:

L 136: Is it possible that ethyl acetate interferes at all?

L 141-143: How does this uncertainty carry through to the conclusions you draw in this study? Since most of your analysis is independent of IEPOX mass, these uncertainties may not affect any major conclusions, but greater discussion of the uncertainties associated with the I- CIMS measurements is warranted. What is the potential for differences in sensitivities to the various compounds measured herein? Is the sensitivity to any given compound known to be constant over the course of a thermogram?

L 144: What were the concentrations of ammonium bisulfate and sulfuric acid, and what size particles did their atomization generate?

L 211-214: This sentence is confusing and may be missing a verb or clause.

L 216: Speaking of vapor wall losses, how might the wall loss of the re-evaporated 2-methyltetrols affect the results of this study? If you have a large sink to the walls, might Henry's law equilibrium effectively pull more tetrol out of the particles that would otherwise occur?

Figure 4: Should there be units on UHP N2?

L 335: Is the assumption of no particle-phase diffusion limitations a safe one? How might phase separation or organic coatings change these estimates?

L 339-342: How close do you expect these estimates of the Henry's law constant to be? A variation of two orders of magnitude seems strikingly large.

L 460: Since other reported values (e.g. Figure 8) are in ug/m3, it would be helpful to report the same units for the starting IEPOX concentration

L 460-461: Why were 90% and 10% chosen? How certain are these branchings, and how wide a range might they span in ambient conditions? How sensitive are the simulation results to changes in these numbers, and in the other parameters used?

L 463-464: What rates are used for the gas-phase reactions with OH?

L 499: Hyphen in the wrong place on "2 methyl-tetrol"?

Reference:

Hu, W., Palm, B. B., Day, D. A., Campuzano-Jost, P., Krechmer, J. E., Peng, Z., de Sá, S. S., Martin, S. T., Alexander, M. L., Baumann, K., Hacker, L., Kiendler-Scharr, A., Koss, A. R., de Gouw, J. A., Goldstein, A. H., Seco, R., Sjostedt, S. J., Park, J.-H., Guenther, A. B., Kim, S., Canonaco, F., Prévôt, A. S. H., Brune, W. H., and Jimenez, J. L.: Volatility and lifetime against OH heterogeneous reaction of ambient isoprene-epoxydiols-derived secondary organic aerosol (IEPOX-SOA), Atmos. Chem. Phys., 16, 11563-11580, https://doi.org/10.5194/acp-16-11563-2016, 2016.

Lopez-Hilfiker, F. D., Mohr, C., D'Ambro, E. L., Lutz, A., Riedel, T. P., Gaston, C. J.,
Iyer, S., Zhang, Z., Gold, A., Surratt, J. D., Lee, B. H., Kurten, T., Hu, W. W., Jimenez, J., Hallquist, M., and Thornton, J. A.: Molecular composition and volatility of organic aerosol in the southeastern US: Implications for IEPOX derived SOA, Environ. Sci. Technol., 50, 2200-2209, doi: 10.1021/acs.est.5b04769, 2016.
* * *

---

## Author Comment (AC1) · 26 Jun 2019

Comments prepared by Lindsay Yee, Gabriel Isaacman-VanWertz, and Allen Goldstein

We thank the commenters for their careful reading of our manuscript and helpful comments. Our point-by-point responses are below the comments, highlighted in blue.

Most of the comments by Dr. Yee et al. are centered around how the FIGAERO-CIMS measurements described in our manuscript relate to specific chemical compounds reported by other measurement techniques (e.g. GC/MS), especially given that the FIGAERO-CIMS does not separate by structure/isomer. We think it is therefore helpful to start by summarizing some aspects of FIGAERO-CIMS as a top-down constraint on molecular components and how those inform our main scientific conclusions in this study.

Our goal in this study is not to explain the measurement of specific compounds, such as $C_5$-alkene triols reported by other instruments, but rather to draw conclusions about the presence of $C_5$-alkene triols and related isomers generally in ambient SOA and specifically in SOA generated from reactive uptake of IEPOX. We have clarified that our work in no way comments on the potential for $C_5$-alkene triols to be present in the gas-phase, nor that other measurement techniques may measure $C_5$-alkene triols from sources other than IEPOX SOA. However, as we discuss below, if $C_5$-alkene triols and related isomers are in SOA, then the FIAGERO-CIMS would measure them, and this constraint provides insight into the nature of $C_5$-alkene triols, $C_5$-furan diols, and related isomers in ambient SOA and IEPOX-derived SOA specifically.

We do not have any reason to expect that the FIGAERO-CIMS using Iodide adduct ionization is not sensitive to $C_5$-alkene triols, and furan diols, based on direct calibrations of compounds with nearly identical structures shown in part in this manuscript and others [*Iyer et al.*, 2016; *Lopez-Hilfiker et al.*, 2016]. Thus, we fully expect to measure $C_5$-alkene triols and furan diols [*Iyer et al.*, 2016] should they exist in the particle phase. While FIGAERO-CIMS does not speciate across isomers, the signal at $C_5H_{10}O_3$ is a top-down constraint on the sum-total of $C_5$-alkene triols, furan diols, hydroxy hydroperoxides, dihydroxy epoxides, hydroxy carboxylic acids, and dihydroxy carbonyls.

Importantly, we have yet to find a structure with the $C_5H_{10}O_3$ composition that is not detected by FIGAERO-CIMS. We have calibrated to IEPOX, ISOPOOH, alkane triols, and a furan diol with this specific composition, as well as similar hydroxy acids and hydroxy carbonyls [*Lee et al.*, 2014]. Thus, the behavior of $C_5H_{10}O_3$ components in the FIGAERO-CIMS is a constraint on the behavior of commonly measured IEPOX SOA tracers. We think it useful to make this point in the manuscript, that commonly measured tracers of IEPOX SOA reported by other techniques, such as the $C_5$-alkene triols and furan diols, would fall into this category and be measured by FIGAERO-CIMS.

Summarizing the above, if $C_5$-alkene triols, furan diols, IEPOX, or ISOPOOH are present in the aerosol, the $C_5H_{10}O_3$ signal in the FIGAERO-CIMS accounts for that carbon. We likely underestimate the amount of furan diols due to a slightly higher sensitivity to triols, on which we base our conversion of $C_5H_{10}O_3$ signal to mass concentrations and is a reason we have a low-bias compared to the AMS. That is, stating that the FIGAERO-CIMS accounts for ~50% of IEPOX PMF factor mass from the AMS is a conservative estimate because we currently underestimate contributions from furan diols. It is also possible that the AMS factor is not completely specific to IEPOX derived SOA.

Given that i) we do not observe multiple or broad desorption profiles of $C_5H_{10}O_3$ (which would possibly imply multiple isomers), ii) that we replicate the field measured desorption profile of $C_5H_{10}O_3$ in the laboratory using only IEPOX reactive uptake, and iii) that the $C_5H_{10}O_3$ desorption profile is only consistent with a species having much lower volatility than that of a freely partitioning $C_5$-alkene triol or furan diol (or IEPOX, ISOPOOH, etc), we conclude that such isomers do not exist in organic aerosol

freely and are instead artifacts of thermal decomposition of a much lower volatility product such as organosulfates, or associated IEPOX-derived oligomers, known independently to also form from IEPOX reactive uptake.

If such components of IEPOX SOA thermally decompose in the FIGAERO, a reasonable hypothesis is that they thermally decompose in other analytical methods that also similarly heat the sample, potentially leading to the same detected tracers ($C_5H_{10}O_3$ isomers). We have reframed our conclusions to make clear that this extrapolation to other methods is a hypothesis that can be tested, and that to some degree has been, and is supported by independent experiments by Cui et al. [2018]. More direct comparisons could certainly provide better tests of specific isomers desorbing from IEPOX SOA or other types of organic in different analytical methods.

In light of the uncertainty surrounding the molecular structure(s) that actually lead to observations of C5-alkene triols in chromatography based techniques and C5H10O3 measurements in CIMS, we think that there are areas of your manuscript that should be adjusted per below:

I.       Citation of literature:

      a.   Line 87: Reference to Isaacman-VanWertz et al., ES&T 2016 would be appropriate here

Reference added.

      b.   Section starting line 289: Why not also compare results to field measurements of volatility as in Hu et al, ACP, 2016, DOI: 10.5194/acp-16-11563-2016?

We have added a reference to Hu et al. [2016] on line 377 when discussing the volatility of the ambient aerosol.

      c.   Consider providing additional chemical understanding of proposed compounds in Figure 7 and their expected behavior under programmed thermal desorption and hydrolysis affecting oligomer/accretion products recovery as in Claflin and Ziemann et al., AS&T, 2019, DOI: 10.1080/02786826.2019.1576853.  Are your isothermal evaporation results with humidified air providing opportunity for hydrolysis?  While lines 52-56 state that your methods confirm that these products are artifacts of thermal decomposition/hydrolysis, hydrolysis is not discussed explicitly elsewhere in the manuscript as it applies to your measurements.

For Figure 7, we are highlighting possible pathways that could explain the time-evolution of our measurements (Figure 6), and effective volatility of IEPOX SOA components that we measured. We have added a statement to this effect on lines 430-432 (copied below) and changed the figure caption to highlight that these are hypothetical processes, albeit based on evidence in other publications. As for hydrolysis, we are evaporating the aerosol at the same RH as their formation, i.e. 50% RH, so we do not believe we would be driving more hydrolysis than would be occurring in the chamber. We have removed the statement about hydrolysis from the abstract.

*"...illustrated in Figure 7 which shows hypothetical reaction pathways and oligomers that could explain the observed time-evolution of detected products."*

      d.   Lines 224-227: Reword to be more accurate.  Lopez-Hilfiker et al., 2016, pg. 2204 states that a sum of tracers in CIMS measurement that are highly correlated with C5H12O4 and C5H10O3 explain 50% of IEPOX-SOA factor mass, not just C5H12O4 and C5H10O3.

Figure 1 in Lopez-Hilfiker et al. [2016] shows that the sum of FIGAERO-CIMS tracers varies from explaining ~50-90% of the AMS IEPOX-PMF factor. Figure S3 shows that $C_5H_{10}O_3$ + $C_5H_{12}O_4$ make up >80% of the total sum of tracers (which includes 6 additional ions). Thus our wording on lines 227-230 that the FIGAERO-CIMS $C_5H_{10}O_3$ + $C_5H_{12}O_4$ make up ~50% of IEPOX SOA mass measured by the AMS is accurate with reference to Lopez-Hilfiker et al. [2016]. See also above discussion of the IEPOX SOA reported by the FIGAERO-CIMS in Lopez-Hilfiker et al. [2016] being a conservative estimate.

     e.   Line 455: Do you mean Lin et al., ES&T 2014, DOI: 10.1021/es503142b, instead of Lin et al., ES&T 2012, or in addition?

We have added a reference to Lin et al., ES&T 2014 in addition.

    II.   Be more precise in language throughout the manuscript relating instrument's observation of a chemical formula (which even stated in line 47 that C5H12O4 is "presumably" 2-methyltetrols) yet in other areas claim unwarranted certainty in the chemical structure of C5H12O4 and C5H10O3. This also needs to be adjusted regarding relation to findings under chamber conditions and then related to atmosphere. Language should be adjusted throughout; some examples specifically listed below:

     a.   Lines 52-56: "We thus confirm, using controlled laboratory studies, recent analyses of ambient SOA measurements showing that IEPOX SOA is of very low volatility and commonly measured IEPOX SOA tracers, such as methyltetrols and C5-alkene triols, result predominantly from artifacts of measurement techniques associated with thermal decomposition and/or hydrolysis." As it is not verified if the C5-alkene triol structure is what gives C5H10O3 in CIMS (standards don't exist yet, though semi-reasonable to assume from a chamber experiment), this statement is not supported enough. Further, Surratt's HILIC method (Cui et al., Environ. Sci. Process. Impacts, 2018) which does measure some higher order organosulfates and oligomers still does not account for all C5-alkene triols as stemming from decompositon (<50%), so is it fair to say predominant? Maybe just predominant for chamber conditions tested here?

As discussed above, our measurement of $C_5H_{10}O_3$ would include $C_5$-alkene triols and 3-MeTHF-3,4-diols, as well as other related isomers. We have changed the wording throughout the manuscript to use the measured formula for the tracers and indicate that they are presumably, but not definitively, the 2-methyltetrols and $C_5$-alkene triols or 3-MeTHF-3,4-diols. We can't think of another structure other than the 2-methyltetrols for $C_5H_{12}O_4$ appearing from authentic IEPOX reacting in the dark in the presence of only aqueous acidic aerosol and no oxidants. We'd happily add a reference to other isomers that would have that composition under those conditions should we be made aware of them.

     b.   Lines 220-224: The chemical formulae observed by FIGAERO CIMS are consistent with those from actual chemical species observed by other techniques. Line 221 should be adjusted to something more like, "These chemical formulae are consistent with those of chemical species (i.e. 2-methyltetrols and C5-alkene triols, respectively) that have been repeatedly shown to be major components of IEPOX SOA…"

We have changed the wording to now read: "Species with these compositions have been repeatedly shown to be major components of IEPOX SOA".

     c.   While lines 270-278 are precisely worded to state that the interpretation of C5H10O3 as observed in chamber SOA derives primarily from thermal degradation, this does not warrant broad generalization to the atmosphere as in lines 52-56, lines 494-

497.  Considering there are other potential chemical conditions under which C5-alkene triols may be observed, but not tested here there should be room left for these possibilities, even if less abundant in the atmosphere including:

We basically agree that there are likely multiple sources of $C_5H_{10}O_3$ during a thermal desorption and this is what we're trying to convey in figure 7. While the isothermal evaporations are evidence that $C_5H_{12}O_4$ exists partly as a monomer in the particle phase, we don't have any evidence that isomers of $C_5H_{10}O_3$ from IEPOX uptake exists in a monomer form in the particle phase. Due to the simplicity of the inputs (IEPOX + acidified ammonium sulfate + aqueous seed), there are limited combinations of monomeric units possible in these chamber experiments, hence why two species ($C_5H_{12}O_4$ & $C_5H_{10}O_3$) make up such a large portion of our measurements. In addition, the fact that our chamber and ambient thermograms for $C_5H_{10}O_3$ ([*Lopez-Hilfiker et al.*, 2016], figure 1) are rather narrow suggests that what we are measuring also likely comes from compound/s that have similar bond strengths that are breaking within a fairly narrow temperature range to produce $C_5H_{10}O_3$ (see also Shobesberger et al. [2018]). As noted above, these components have nearly the same effective volatility ($T_{max}$) as measured by the FIGAERO in chamber generated IEPOX SOA and in ambient OA measured in the field. Thus, the field measurements made with the same instrument and setup [*Lopez-Hilfiker et al.*, 2016], are very similar to the chamber results, and this similarity is the support for concluding more generally about the sources of $C_5H_{10}O_3$ in IEPOX SOA.

    i.   IEPOX uptake on non-acidified seed/dry particles:

        1.   Nguyen et al., ACP, 2014, DOI: 10.5194/acp-14-3497-2014

        2.   Riva et al., ES&T 2016, DOI: 10.1021/acs.est.5b06050 (Fig. 2 shows C5-alkene triols observed under dry conditions and wet conditions)

        3.   Lin et al., ES&T 2014, DOI: 10.1021/es503142b (Fig. S2 shows C5-alkene triols observations under wet_netural and dry_acidic conditions)

        4.   D'Ambro et al., ACP, 2017, DOI: 10.5194/acp-17-159-2017 (Fig. 4 shows C5H10O3 observation)

We respectfully disagree that the above references represent counter examples to our conclusions. IEPOX can undergo the same ring-opening chemistry and oligomerization in aqueous ammonium sulfate particles which likely have a pH of 4-5 based on aerosol thermodynamic modeling, and will depend on RH [*Gaston et al.*, 2014], only slower. In references 2 and 3, "dry" does not necessarily mean "solid" as the acidification used likely leads to ammonium bi sulfate or even sulfuric acid solutions which remain deliquesced even if some ammonium sulfate effloresces at low RH. We show below that the $C_5H_{10}O_3$ reported in reference 4, which did use effloresced ammonium sulfate seed, is present at nearly 2 orders of magnitude lower mass fraction of the OA with a desorption profile markedly different than the IEPOX tracer discussed in this work.

[Figure]

ii. Non-IEPOX pathways:

    1. Liu et al., ES&T 2016, DOI: 10.1021/acs.est.6b01872, as C5H10O3 is also observed here reported in Figure 3

    2. Riva et al., ES&T 2016, DOI: 10.1021/acs.est.6b02511, as C5-alkene triols are observed as reported in Figure 1

iii. isoprene ozonolysis leading to structures proposed as sulfate esters of C5-alkene triols as in Riva et al., Atmos. Env. 2016, DOI: 10.1016/j.atmosenv.2015.06.027)

As noted above, we agree that $C_5H_{10}O_3$ components could arise from other pathways. But, reference 1 under non-IEPOX pathways is based on the same set of experiments as in reference 4 above, and as noted the $C_5H_{10}O_3$ in that system behaves nothing like what we describe for the IEPOX system nor what we observe in ambient SOA. As for isoprene ozonolysis leading to sulfate esters, we cannot comment as we have not conducted similar experiments, but our ambient measurements would suggest they either decompose in the FIGAERO the same as IEPOX-derived organosulfates or they are a small contribution to ambient SOA.

III. The connection of FIGAERO-CIMS measurement of C5H10O3 being equivalent to the same C5-alkene triols signal reported from GC-MS is not clear unless an intercomparison was done previously. If done, then please cite the reference:

a. Lines 494-497: Claim of confirmation that C5-alkene triols and/or 3-MeTHF-3,4-diols are all artifacts of thermal decomposition is too strong given that it has not been established if what the FIGAERO CIMS measures as these compounds (C5H10O3) is actually the same as what other techniques (GC/MS) would assign to be C5-alkene triols and/or 3-MeTHF-3,4-diols. Where is there direct comparison of FIGAERO-CIMS measured C5H10O3 and other techniques measured C5-alkene triols and/or 3-MeTHF-3,4-diols to prove that the compound/s is/are the same?

We refer to the general discussion above about the top-down constraint provided on these isomers with the FIGAERO CIMS. We have not attempted direct comparisons but agree such could be useful. However, we don't think such comparisons are required to make our conclusions given that the FIGAERO CIMS is sensitive to these isomers, and that using similar or even authentic standards of them demonstrates how different the IEPOX SOA and ambient OA components with the same composition behave during a thermal desorption. For example, we directly compare the thermal desorption of pure 3-MeTHF-3,4-diols to thermal desorptions of the IEPOX SOA components and show they are entirely

inconsistent with these isomers being present as free monomers in the aerosol. Our best explanation is thermal decomposition. The new experiments described here illustrate that another possible explanation, monomers trapped by highly viscous OA, is unlikely due to the rapid and nearly complete evaporation of the species measured at $C_5H_{12}O_4$.

      b. Further, this means the conclusive statement in lines 522-523 is not supported to broadly overgeneralize that GC/MS methods are incorrect. As there are a variety of ways in which GC/MS methods are employed to measure isoprene-derived tracers as well as how the samples are handled and treated before GC/MS analysis, it does not seem appropriate to mention this here without performing proper intercomparisons or going into more specifics as to why all GC/MS methods would lead to thermal decomposition of these products. Thus, statements about specific tracers being "artifacts" or somehow not being useful measurements are misleading and unnecessary. It seems more productive to focus on what the community can learn from the different methods, and what the different tracers teach us, rather than denigrating a particular measurement approach because it sees a combination of products that may include pieces that decompose from larger molecules. For example, the UPLC/HILIC/GC-MS methods provide more specific information (more highly speciated EI mass spectrum of individual tracers that elute from chromatography columns) than the direct CIMS (sum of isomers as a function of desorption temperature) or AMS (EI mass spectrum of the total), but using all the observations we can learn more (not less) about the chemical composition, sources, and transformation processes in organic aerosols.

We stress that our goal is not to denigrate any particular instrument or method. Moreover, we further stress that even though we conclude $C_5H_{10}O_3$ components are detected in the FIGAERO CIMS analysis of IEPOX SOA due to thermal decomposition, those tracers are highly useful for source apportionment of OA. And we agree that the suite of instruments are useful and necessary for making progress on understanding the sources and properties of SOA.

We have made it clearer that our results suggest a hypothesis that other methods that utilize heat in the workup procedures or online analyses would detect IEPOX SOA tracers which are the result of thermal decomposition of low volatility components. We also make it clearer that, using the FIGAERO CIMS specifically, we find no evidence for IEPOX SOA components with an elemental composition of $C_5H_{10}O_3$ and desorption profile consistent with a free monomer partitioning based on its solubility or saturation vapor pressure. The resulting section, which also address points from IIIa above, are on lines 516-523 and below.

*We further confirm that the observed properties of $C_5H_{10}O_3$ are not consistent with the structure of $C_5$-alkene triols and/or 3-MeTHF-3,4-diols, and thus these structures cannot be components of IEPOX SOA but are likely artifacts of thermal decomposition during analytical workup. A direct intercomparison is required to definitively determine whether all instruments are measuring the same species and that prior estimates of IEPOX SOA have not been overestimated due to "double counting" carbon in these tracers which might be derived from organosulfates and oligomers measured separately.*

*The evidence presented herein, as well as in independent experiments [Cui et al., 2018], indicates that the $C_5H_{10}O_3$, regardless of structure, as well as a significant portion of the $C_5H_{12}O_4$, are not actual components of the SOA but rather derived from other related components during the analysis. Therefore, we do not recommend that these species be included as products in mechanistic models of IEPOX SOA formation and evolution.*

IV.     For some of the structures (organosulfates/polymers) proposed to lead to C5H10O3 via decomposition, can you estimate their C*/Tmax and do they make sense with the observed thermograms?

We do not have direct measurements of the species that decompose and lead to the detection of $C_5H_{10}O_3$ as they do not survive our analysis. However, we can estimate the c* of the dimer species in Figure 7 based on group contribution methods [*Compernolle et al.*, 2011]. The very low c* supports the idea that these species with their weaker ether and sulfate bridges will likely undergo decomposition prior to volatilization.

| Structure | SMILES | Estimated c* ($\mu g/m^3$) |
|---|---|---|
|  | CC(CO)(OCC(O)C(C)(CO)OS(O)(=O)= O)C(O)CO | 0.0001 |
|  | CC(O)(CO)C(O)COS(=O)(=O)OC(CO)C(C)(O)CO | 0.00006 |

**References**

Compernolle, S., K. Ceulemans, and J. F. Muller (2011), EVAPORATION: a new vapour pressure estimation method for organic molecules including non-additivity and intramolecular interactions, *Atmos. Chem. Phys.*, *11*(18), 9431-9450.

Cui, T., et al. (2018), Development of a hydrophilic interaction liquid chromatography (HILIC) method for the chemical characterization of water-soluble isoprene epoxydiol (IEPOX)-derived secondary organic aerosol, *Environmental Science: Processes & Impacts*.

Gaston, C. J., T. P. Riedel, Z. F. Zhang, A. Gold, J. D. Surratt, and J. A. Thornton (2014), Reactive uptake of an isoprene-derived epoxydiol to submicron aerosol particles, *Environ. Sci. Technol.*, *48*(19), 11178-11186.

Hu, W. W., et al. (2016), Volatility and lifetime against OH heterogeneous reaction of ambient isoprene-epoxydiols-derived secondary organic aerosol (IEPOX-SOA), *Atmos. Chem. Phys.*, *16*(18), 11563-11580.

Iyer, S., F. Lopez-Hilfiker, B. H. Lee, J. A. Thornton, and T. Kurten (2016), Modeling the Detection of Organic and Inorganic Compounds Using Iodide-Based Chemical Ionization, *Journal of Physical Chemistry A*, *120*(4), 576-587.

Lee, B. H., F. D. Lopez-Hilfiker, C. Mohr, T. Kurten, D. R. Worsnop, and J. A. Thornton (2014), An iodide-adduct high-resolution time-of-flight chemical-ionization mass spectrometer: application to atmospheric inorganic and organic compounds, *Environ. Sci. Technol.*, *48*(11), 6309-6317.

Lopez-Hilfiker, F. D., et al. (2016), Molecular composition and volatility of organic aerosol in the southeastern US: Implications for IEPOX derived SOA, *Environ. Sci. Technol.*, *50*(5), 2200-2209.

Schobesberger, S., E. L. D'Ambro, F. D. Lopez-Hilfiker, C. Mohr, and J. A. Thornton (2018), A model framework to retrieve thermodynamic and kinetic properties of organic aerosol from composition-resolved thermal desorption measurements, *Atmos. Chem. Phys.*, *18*(20), 14757-14785.

---

## Author Comment (AC2)

We thank the referees for their valuable comments. We respond below each comment in blue highlighted text and indicate the corresponding changes to the manuscript where relevant.

Anonymous Referee #1

General comments

This manuscript describes laboratory experiments using the Filter Inlet for Gases and Aerosols/Chemical Ionization Mass Spectrometry (FIGAERO-CIMS) technique that aim to investigate the nature of the components of IEPOX-derived secondary organic aerosol (IEPOX-SOA). Specifically, the work addresses the inconsistency between GCMS approaches that have identified semi-volatile molecular components and volatility measurements that have indicated that the bulk of IEPOX-SOA must be made up of much lower volatility molecular components. The main claim is that a desorption signal that corresponds to C5H12O4 has two maxima, one that arises from a semi-volatile source (presumably from 2-methyl tetrols directly evaporating) and one that arises from the thermal decomposition of a low volatility source and a desorption signal that corresponds to C5H10O3, which also arises from the thermal decomposition of a low volatility source. Because knowledge of the molecular composition of IEPOX-SOA is critical to the development of accurate SOA mechanistic models, this work will be of great interest to readers of Atmospheric Chemistry and Physics. However, I believe that a number of uncertainties remain in the interpretation of both the present work and past studies, and that a revised manuscript should more directly address these issues.

Specific comments

Because of a lack of authentic standards, there continues to be no proof whatsoever that either the GC/MS signals previously attributed to C5-alkene triols or the present CIMS signals attributed to C5-alkene triols actually correspond to these species. Indeed, Watanabe et al. 2018 showed that these species are among the least thermodynamically favored among a variety of possible C5H10O3 isomers. I suggest that the manuscript be revised to simply refer to a C5H10O3 thermal decomposition product and refrain from associating this product with any particular molecular form.

This is a good point. We have changed and clarified wording, specifically on lines 123-125, 294, and 511, however we do think it useful in various places to connect this component to tracers of IEPOX SOA reported in the literature with the same elemental composition.

Similarly, while I find the argument fairly convincing that the semi-volatile C5H12O4 component is probably the 2-methyl tetrols themselves, I don't think the low volatility thermal decomposition product can be assumed to be the 2-methyl tetrols.

Agree, our point was supposed to be that there is a semi-volatile component with the composition $C_5H_{12}O_4$, likely the tetrol, but also an additional lower volatility component that decomposes into $C_5H_{12}O_4$ during the thermal desorption as reported in field measurements by Lopez-Hilfiker et al. [2016]. We have adjusted the wording on lines 294, and 337-338.

The proposed oligomerization mechanism given in Figure 7 would benefit from more detailed discussion. Rather than a more obvious mechanism directly involving IEPOX, the authors are proposing two types of reactions: 1) acid-catalyzed etherification reactions of organosulfates to form ether-linked oligomers and 2) acid-catalyzed sulfate esterification reactions to form sulfate-linked oligomers. The

authors don't provide any literature precedents for these types of reactions. Therefore, I think the authors should provide a rationale for these somewhat unusual reaction types.

We have clarified that the schematic in Figure 7 is a hypothetical set of net transformations to explain the slower shifts in IEPOX SOA volatility and thermogram shape presented in Figure 6. We think a majority of the IEPOX SOA is low volatility, and agree most is indeed formed promptly through IEPOX aqueous phase chemistry to form the organosulfate, or even polyols. These other hypothetical reactions are to offer an explanation for the slower evolution of IEPOX-derived SOA volatility/composition taking place in the absence of IEPOX during isothermal evaporation experiments and at longer times in the batch mode experiments when prompt SOA formation has slowed or even ceased but the SOA continued to age.

Along the same lines, the overall interpretation would benefit from estimates of the thermal desorption behavior of the proposed oligomer components. Did the authors suggest two isomers of a monosulfate dimer as their major proposed molecular species because C10H22SO10 is expected to have roughly the observed thermal desorption behavior?

As stated above, we are unfortunately not able to measure these oligomer (or monomer oganosulfate) species as we suspect they are decomposing completely during our analysis. In our response to Yee et al., the final comment, we have estimated c* based on the EVAPORATION group contribution method [*Compernolle et al.*, 2011] for some of these hypothetical structures and find that they are exceedingly low and thus subject to thermal decomposition during the desorption process before evaporation rates would be sufficient to desorb them from the filter in detectable amounts. We refer the reviewer to other work by the Surratt group proposing and observing the presence of polyol esters [*Lin et al.*, 2014; *Surratt et al.*, 2006; *Zhang et al.*, 2011].

Technical comments

Line 87: There should be a reference to Hu et al. 2016 ACP 16, 11563-11580 here.

Reference added.

Line 128: There is something wrong with the grammar in this sentence. Please revise.

We changed the sentence to:

*The data presented herein were taken at the Pacific Northwest National Laboratory (PNNL) as part of the Secondary Organic Aerosol Formation from Forest Emissions Experiments (SOAFFEE) campaign held during the summer of 2015.*

Line 464: I'm not sure why there is a reference to Atkinson, 1987 here. There are two measurements of IEPOX + OH rate constants: Bates et al. 2014 and Jacobs et al. 2013, ES&T 47, 12868-12876, which differ by a factor of two. The choice of rate constant should be explicitly discussed.

Thank you for pointing this out. The model is very simple so we use one gas-phase OH reaction rate which applies to both the IEPOX and 2-methyltetrol. We used the Atkinson SAR method to determine the OH reaction rate with both 2-methyltetrol and IEPOX. The rate found by the Atkinson method was within the range of values found by Bates et al. and Jacobs et al. We have added the references for

Bates & Jacobs and reworded the sentence (see below) to describe our rate-selection process more clearly and now also include the rate.

*We also include a loss of gas-phase species (2-methyltetrol and IEPOX) due to reaction with OH at $1.8\times10^{-11}$ cm$^3$ molec$^{-1}$ s$^{-1}$ [Atkinson, 1987], consistent with previous studies for the IEPOX + OH rate constant [Bates et al., 2014; Jacobs et al., 2013].*

Anonymous Referee #2

Overview:

This study sheds new light on the question of volatility and chemical composition of secondary organic aerosol derived from IEPOX, a ubiquitous biogenic aerosol component. IEPOX is known to react in aqueous aerosol to form commonly observed products such as methyltetrols and organosulfates, and compounds with the molecular formula $C_5H_{10}O_3$ (C5 alkene triols, methyl-tetrahydrofuran-diols, or both) have also been observed in IEPOX SOA. The total reactivity of IEPOX in the aerosol phase and its uptake are known to be highly dependent on aerosol acidity, sulfate content, organic coating, and other parameters, but a detailed chemical understanding of the composition of the resulting organic aerosol is still lacking. Most observations of ambient IEPOX-derived SOA suggest that the majority of the SOA is of lower volatility than the individual species described above, suggesting a greater presence of oligomers and organosulfates and relatively small contributions from methyltetrols, alkene triols, or MeTHF-diols. This work bridges that apparent gap by showing that observed alkene triols or MeTHF-diols are artifacts formed during the thermal decomposition of lower volatility material, as is a portion of the signal observed at the same mass as methyltetrols, while another portion that likely arises from the methyltetrols themselves is volatile and undergoes evaporation from the particle phase on the timescale of 1 hour.

The authors reach these conclusions by observing the uptake of trans-b-IEPOX onto acidified ammonium bisulfate seed aerosol in a series of controlled chamber experiments. They evaluate the evolution of particle chemical composition and component volatility using a high-resolution time-of-flight I- chemical ionization mass spectrometer outfitted with a filter inlet for gases and aerosols, which enables the separation of isobaric compounds by thermal desorption. Thermograms taken at varying humidities and timepoints during the experiments support the assertion that methyltetrols form quickly via direct hydrolysis following IEPOX uptake, and are then both transformed to lower-volatility species and evaporated into the gas phase, where they can be lost to walls or further photooxidation. The remaining IEPOX-derived SOA is therefore of very low volatility, likely including oligomers of the tetrols and organosulfates. Finally, the authors use a simple box model to illustrate how these results might play out under ambient conditions, and show that models using either aqueous uptake or volatility basis set schemes should be modified to account for the high uptake probability of IEPOX followed by revolatilization of methyltetrols.

While the description of the experiments performed herein and the conclusions drawn from thermal desorption measurements is straightforward and represents a valuable contribution to our understanding of the volatility and chemical composition of IEPOX derived SOA, the extrapolation to ambient conditions remains tenuous and deserves further attention. Additional comparison to ambient results (e.g. Hu et al 2016) would help convince the reader that the results from these chamber studies are borne out in the atmosphere and are relevant to processes that occur in isoprene- and sulfate rich settings. The brief comparison on lines 221-232 is not sufficient, and if anything brings up more questions than it answers as to the validity of comparing field to laboratory results. Does Lopez-Hilfiker et al. suggest that a further 50% of IEPOX SOA observed in ambient consists of tracers that don't even correlate with those identified here, in addition to the fraction that correlates with them but isn't explicitly $C_5H_{12}O_4$ or $C_5H_{10}O_3$? What are those other tracers and what could explain their absence in these experiments? Ambient particles likely contain much more variability in organic compounds in the

condensed phase with which tetrols and other IEPOX-derived species might oligomerize or otherwise interact. How would that affect the conclusions here and our ability to put simple parameterizations into models? Is there any evidence that products of such cross-reactions result in similar observed compounds upon thermal decomposition?

See the response to comments by Yee et al., opening statement and section II c. To summarize point by point here:

1. We further discuss the ambient measurements on lines 281-289, and now cite Hu et al. 2016 on lines 89-90 and 377.
2. Due to the underestimated contribution from furan diols, the stated 50% explanation of the IEPOX PMF factor mass from the AMS is a conservative underestimate which is discussed within the reference.
3. We agree, with more complexity in the atmosphere it is possible and even likely that other species will oligomerize with IEPOX itself or its reaction products, leading to a more complex picture discussed herein. If these monomeric units oligomerize with different species in the atmosphere, forming different linkages, this will affect their lifetime. The different linkages may also affect the compounds observed upon thermal desorption as noted. However, we measure these monomers in the atmosphere, as have numerous other studies, and rather few other components in our mass spectra correlate with these specific tracers over time, suggesting that the products of a broader set of cross reactions either decompose into the same monomeric composition, are not significant contributors, or are for some reason not strongly correlated with the other IEPOX tracers. However, this complexity does not change the main conclusions of our work or implications for modeling, mainly that IEPOX SOA components with the formula $C_5H_{12}O_4$ likely represent a major prompt product of IEPOX reactive uptake, but are semi-volatile and thus can partition to the gas-phase, and otherwise remaining components detected as $C_5H_{12}O_4$ or $C_5H_{10}O_3$ in the particle phase are likely decomposition of lower volatility oligomers or organosulfates.

Technical comments:

L 136: Is it possible that ethyl acetate interferes at all?

We do not have evidence that it is taken up into the aerosol and would not expect it to as its volatility is very high and solubility rather low. In the continuous flow experiments, the ethyl acetate evaporated completely well before the experiment was over and IEPOX was still present in the chamber. Also, given the similarity between the ambient tracers and chamber SOA composition, it would seem there isn't a major effect.

L 141-143: How does this uncertainty carry through to the conclusions you draw in this study? Since most of your analysis is independent of IEPOX mass, these uncertainties may not affect any major conclusions, but greater discussion of the uncertainties associated with the I- CIMS measurements is warranted. What is the potential for differences in sensitivities to the various compounds measured herein? Is the sensitivity to any given compound known to be constant over the course of a thermogram?

Please see our response to Yee et al. for more detail. Briefly: the I-CIMS is able to measure the species of interest here, namely $C_5H_{10}O_3$ and $C_5H_{12}O_4$, regardless of structure [*Iyer et al.*, 2016]. However, whether the $C_5H_{10}O_3$ is an alkene triol or furan diol will affect the sensitivity and thus estimates of mass concentration. We use calibrations to authentic triols and diols to constrain these estimates, and generally use instrument response to $C_5$ triols which is higher per ng than it is for the diols so that our mass concentration estimates are lower limits. Calibration uncertainties are typically 30-50% based on repeated tests, and usually systematic in nature because precision within any individual calibration test is high. We primarily focus on qualitative behavior or intrinsic relative behaviors (e.g. $T_{max}$) in this paper, and thus relative sensitivities do not affect our conclusions.

L 144: What were the concentrations of ammonium bisulfate and sulfuric acid, and what size particles did their atomization generate?

We have added this information in lines 148-157, and copied below.

*Wet, polydisperse acidic ammonium sulfate seed was generated by atomizing an ammonium bisulfate solution acidified with additional $H_2SO_4$. The solution was made by mixing ammonium sulfate (0.1308 g) with sulfuric acid (8.02 mL of 0.2465 M) and diluting to a total volume of 1 L with ultrapure water. The average molar $NH_4^+:SO_4^{2-}$ ratio measured by the AMS was approximately 0.93 for all experiments, though due to the experimental procedure some interference from organic sulfate formation may exist. The measured $NH_4^+:SO_4^{2-}$ ratio is significantly higher than was present in the atomized solution, implying that excess gas-phase ammonia present in the chamber partially neutralized the injected seed. The seed surface area concentrations were approximately 37,600 and 24,000-27,000 $cm^{-3}$ and the volume weighted mode diameters were 106 and 244-254 nm in continuous and batch modes, respectively. Continuous flow experiments were conducted at 50% RH, while the RH of batch mode experiments was either 30% or 50%.*

L 211-214: This sentence is confusing and may be missing a verb or clause.

Agreed, "when" was changed to "with".

L 216: Speaking of vapor wall losses, how might the wall loss of the re-evaporated 2-methyltetrols affect the results of this study? If you have a large sink to the walls, might Henry's law equilibrium effectively pull more tetrol out of the particles that would otherwise occur?

Indeed, this effect of vapor wall loss is quite possible. Operating in continuous flow mode helps achieve a steady state that mitigates effects of reversible vapor wall loss somewhat. In the batch mode experiments, we do not see a significant loss of the low-$T_{max}$/semi-volatile mode over the experimental time scale. That said, attempts to quantify the IEPOX SOA yield, which was not a goal of this study, do need to consider vapor wall loss of the semi-volatile product. We have added this issue to the implications section when we discuss the evolution of IEPOX SOA and the associated simple box-model (lines 481-490), and copied below.

It is not clear how much vapor wall loss affects the results of this study or if we are underestimating or overestimating IEPOX SOA in the chamber. In the atmosphere, photochemical losses of the semi-volatile tetrol in the gas-phase due to reaction with OH (not present in the chamber) would shift more tetrol out of the particle phase than in our chamber experiments (no OH). Moreover, the much smaller aqueous

Responses to Referees                                                                                          6

volume of ambient aerosol particles in the atmosphere compared to that in the chamber would also lead to more of the tetrol re-partitioning to the gas phase than in the chamber. Thus, while vapor wall loss in the chamber likely lowers particle phase tetrol, that effect might be partially or entirely offset by other processes/conditions which enhance particle phase tetrol in the chamber compared to the atmosphere.

*Vapor wall loss is not considered in the model, which might be resulting in more tetrol evaporating from the particles in our measurements than would occur in the atmosphere. However, operating in continuous flow mode helps to mitigate these issues, and in batch mode we do not observe a significant loss of the low-$T_{max}$/semi-volatile mode. In the atmosphere, photochemical losses of the tetrol and the smaller aqueous volume of the aerosol would lead to partitioning of the tetrol out of the aerosol. Thus, while vapor wall loss in the chamber likely leads to lower particle-phase tetrol, the effect would be offset by these processes in the atmosphere, and so not considering vapor wall loss in the model should not significantly affect our results.*

Figure 4: Should there be units on UHP N2?

No, it is simply indicating when there was and was not flow (i.e. on or off). We added an explanation for this to line 307 and the figure caption.

L 335: Is the assumption of no particle-phase diffusion limitations a safe one? How might phase separation or organic coatings change these estimates?

Due to the rapid (1 hour) and nearly complete evaporation of the low $T_{max}$/high volatility mode of the C5H12O4 species, we have no indication that there is a limitation. Thus, while there may well be organic coatings, we can reasonably conclude there is not a sufficient diffusion limitation to evaporation. The observed evaporation rates are consistent with the expected saturation vapor concentration or solubility calculated or inferred from independent methods.

L 339-342: How close do you expect these estimates of the Henry's law constant to be? A variation of two orders of magnitude seems strikingly large.

The quoted range reflects the range of possible values based on the input uncertainties. The COSMOtherm computation estimates are $4.9 \times 10^8 - 1.1 \times 10^{10}$ M atm$^{-1}$ while the value derived from the observed evaporation rate is more constrained to be $1-2 \times 10^8$ M atm$^{-1}$. As for saturation vapor concentration estimates, we would treat these quantities as order-of-magnitude estimates given the challenges associated with deriving them both computationally and experimentally in the extremes (low vapor pressures or very high solubilities), and for multifunctional compounds where predictions are especially uncertain [*D'Ambro et al.*, 2017]. In the end, $>1 \times 10^8$ M atm$^{-1}$ is probably sufficient for atmospheric modeling.

L 460: Since other reported values (e.g. Figure 8) are in ug/m3, it would be helpful to report the same units for the starting IEPOX concentration

We added the value in ug/m3 after the value in ppb.

L 460-461: Why were 90% and 10% chosen? How certain are these branchings, and how wide a range might they span in ambient conditions? How sensitive are the simulation results to changes in these numbers, and in the other parameters used?

We have made an educated guess based on previous measurements to the branching ratio, and previous measurements of the IEPOX SOA yield, and thus they are not all that certain. Rather, we use the model as stated, for a simple conceptualization of how $C_5H_{12}O_4$ would behave in the atmosphere. It's quite possible that this branching is correct, but may span to 50/50, but likely not much higher than 50% OS production based on previous work. The simulation should not be overly sensitive to the branching ratio as the aerosol concentration quickly converges to being dominated by the OS due to the semi-volatile nature of the tetrol product at a rate mainly determined by the solubility or saturation vapor concentration of the tetrol and its gas-phase loss rates.

L 463-464: What rates are used for the gas-phase reactions with OH?

We have added the rate, $1.8\times10^{-11}$ cm$^3$ molec$^{-1}$ s$^{-1}$, to the manuscript on line 472.

L 499: Hyphen in the wrong place on "2 methyl-tetrol"?

Yes, fixed now.

*Reference:*

Hu, W., Palm, B. B., Day, D. A., Campuzano-Jost, P., Krechmer, J. E., Peng, Z., de Sá, S. S., Martin, S. T., Alexander, M. L., Baumann, K., Hacker, L., Kiendler-Scharr, A., Koss, A. R., de Gouw, J. A., Goldstein, A. H., Seco, R., Sjostedt, S. J., Park, J.-H., Guenther, A. B., Kim, S., Canonaco, F., Prévôt, A. S. H., Brune, W. H., and Jimenez, J. L.: Volatility and lifetime against OH heterogeneous reaction of ambient isopreneepoxydiols-derived secondary organic aerosol (IEPOX-SOA), Atmos. Chem. Phys., 16, 11563-11580, https://doi.org/10.5194/acp-16-11563-2016, 2016.

Lopez-Hilfiker, F. D., Mohr, C., D'Ambro, E. L., Lutz, A., Riedel, T. P., Gaston, C. J., C4 Iyer, S., Zhang, Z., Gold, A., Surratt, J. D., Lee, B. H., Kurten, T., Hu, W. W., Jimenez, J., Hallquist, M., and Thornton, J. A.: Molecular composition and volatility of organic aerosol in the southeastern US: Implications for IEPOX derived SOA, Environ. Sci. Technol., 50, 2200-2209, doi: 10.1021/acs.est.5b04769, 2016.

**References**

Atkinson, R. (1987), A Structure-Activity Relationship for the Estimation of Rate Constants for the Gas-Phase Reactions of OH Radicals with Organic-Compounds *Int. J. Chem. Kinet.*, *19*(9), 799-828.

Bates, K. H., J. D. Crounse, J. M. St Clair, N. B. Bennett, T. B. Nguyen, J. H. Seinfeld, B. M. Stoltz, and P. O. Wennberg (2014), Gas phase production and loss of isoprene epoxydiols, *J. Phys. Chem. A*, *118*(7), 1237-1246.

Compernolle, S., K. Ceulemans, and J. F. Muller (2011), EVAPORATION: a new vapour pressure estimation method for organic molecules including non-additivity and intramolecular interactions, *Atmos. Chem. Phys.*, *11*(18), 9431-9450.

D'Ambro, E. L., et al. (2017), Molecular composition and volatility of isoprene photochemical oxidation secondary organic aerosol under low- and high-NOx conditions, *Atmos. Chem. Phys.*, *17*(1), 159-174.

Iyer, S., F. Lopez-Hilfiker, B. H. Lee, J. A. Thornton, and T. Kurten (2016), Modeling the Detection of Organic and Inorganic Compounds Using Iodide-Based Chemical Ionization, *Journal of Physical Chemistry A*, *120*(4), 576-587.

Jacobs, M. I., A. I. Darer, and M. J. Elrod (2013), Rate Constants and Products of the OH Reaction with Isoprene-Derived Epoxides, *Environmental Science & Technology*, *47*(22), 12868-12876.

Lin, Y. H., H. Budisulistiorini, K. Chu, R. A. Siejack, H. F. Zhang, M. Riva, Z. F. Zhang, A. Gold, K. E. Kautzman, and J. D. Surratt (2014), Light-absorbing oligomer formation in secondary organic aerosol from reactive uptake of isoprene epoxydiols, *Environ. Sci. Technol.*, *48*(20), 12012-12021.

Lopez-Hilfiker, F. D., et al. (2016), Molecular composition and volatility of organic aerosol in the southeastern US: Implications for IEPOX derived SOA, *Environ. Sci. Technol.*, *50*(5), 2200-2209.

Surratt, J. D., et al. (2006), Chemical composition of secondary organic aerosol formed from the photooxidation of isoprene, *Journal of Physical Chemistry A*, *110*(31), 9665-9690.

Zhang, H., J. D. Surratt, Y. H. Lin, J. Bapat, and R. M. Kamens (2011), Effect of relative humidity on SOA formation from isoprene/NO photooxidation: enhancement of 2-methylglyceric acid and its corresponding oligoesters under dry conditions, *Atmos. Chem. Phys.*, *11*(13), 6411-6424.

---

## Author Response (AR2)

We thank the referees for their additional valuable comments clarifying the wording. Our responses are below each comment in blue highlighted text.

Changes in language should still be made to address Referees' comments regarding reference of C5H10O3 associated with C5-alkene triols structures and extrapolation to ambient conditions:

1. Change abstract lines 52-57 to read: "We thus confirm, using controlled laboratory studies, recent analyses of ambient SOA measurements showing that IEPOX SOA is of very low volatility and commonly measured IEPOX SOA tracers such as C5H12O4 and C5H10O3, presumably 2-methyltetrols and C5-alkene triols or 3-MeTHF-3,4-diols, result predominantly from thermal decomposition in FIGAERO-CIMS. We infer that other measurement techniques using thermal desorption or prolonged heating for analysis of SOA components may also lead to reported 2-methyltetrols and C5-alkene triols structures."

Updated as requested.

2. Lines 507-508: Authors have not proven/provided quantitative measurement showing the majority of IEPOX SOA in the experiments and atmosphere are in fact organosulfates/polyol oligomers, though it is suggested and not "confirmed" from modeling. Reword to, "Fundamental chamber studies of IEPOX reactive uptake to aqueous acidic seed were performed and we find that the resulting molecular composition and volatility of the formed SOA suggest that the vast majority of IEPOX SOA in the atmosphere is of very low volatility, likely in the form of organosulfates and polyol oligomers."

Updated as requested.

3. Lines 511-514: This is applicable to FIGAERO-CIMS only. Rewrite as, "We further confirm that the observed properties of C5H10O3 are not consistent with the structure of C5-alkene triols and/or 3-MeTHF-3,4-diols, and thus these structures cannot be components of IEPOX SOA as measured by FIGAERO-CIMS but are likely artifacts of thermal decomposition during analytical workup."

Many techniques measure the composition of $C_5H_{10}O_3$, and while the structure may differ between these techniques, we have shown via numerous methods throughout the paper (structure-activity vapor pressure calculations, calibrations with authentic standards, and modeling with a simple box model, COSMOtherm, and thermal desorption model) that regardless of structure, it is physically inconsistent for that composition to remain in the condensed-phase. We have added the following sentence after the sentence in question to address this comment:

*While our results are specific to the FIGAERO, we predict that the issue is more general, affecting other methods as indicated for example in Cui et al. (2018).*

4. Lines 514-517: Suggest further clarification by changing to, "…whether all instruments (those

reporting C5H12O4, C5H10O3, 2-methyltetrols, C5-alkene triols and/or 3-MeTHF-3,4-diols) are in fact measuring the same species…"

Updated as requested.